# LooGLE v2: Are LLMs Ready for Real World Long Dependency Challenges?

**Ziyuan He**[1,*] **Yuxuan Wang**[3,*], **Jiaqi Li**[2,*], **Kexin Liang**[1,], **Muhan Zhang**[1,2†]

[1]Institute for Artificial Intelligence, Peking University
[2]National Key Laboratory of General Artificial Intelligence, BIGAI
[3]School of Computer Science and Technology, Beijing Institute of Technology
`https://github.com/MuLabPKU/LooGLE-v2`

## Abstract

Large language models (LLMs) are equipped with increasingly extended context windows recently, yet their long context understanding capabilities over long dependency tasks remain fundamentally limited and underexplored. This gap is especially significant in many real-world long-context applications that were rarely benchmarked. In this paper, we introduce **LooGLE v2**, a novel benchmark designed to evaluate LLMs' long context ability in real-world applications and scenarios. Our benchmark consists of automatically collected real-world long texts, ranging from 16k to 2M tokens, encompassing domains in law, finance, game and code. Accordingly, we delicately design 10 types of domain-specific long-dependency tasks and generate 1,934 QA instances with various diversity and complexity in a scalable data curation pipeline for further practical needs. We conduct a comprehensive assessment of 6 locally deployed and 4 API-based LLMs. The evaluation results show that even the best-performing model achieves only a 59.2% overall score on our benchmark. Despite the extensive context windows, popular LLMs are only capable of understanding a much shorter length of context than they claim to be, revealing significant limitations in their ability to handle real-world tasks with long dependencies and highlighting substantial room for model improvement in practical long-context understanding.

## 1 Introduction

In recent years, substantial advancements have been achieved in extending the context length of large language models (LLMs). The context window used during pretraining and post-training has grown dramatically—from 8K to 128K tokens (Dubey et al., 2024; Jiang et al., 2023a; Abdin et al., 2024; Team et al., 2025; Young et al., 2024), and even up to 1M tokens (Yang et al., 2025; Zeng et al., 2024). These advancements have led to improved performance on existing long-context benchmarks. Despite the remarkable increase in context length, current long context benchmarks predominantly focus on document question answering (QA) tasks, emphasizing narrow capabilities like information retrieval and simple reading comprehension from a lengthy document. There exists a significant gap for current LLMs between their ability to process extended input sequences and their practical abilities of resolving real-world problems.

First and foremost, existing long context benchmarks fail to address the diverse, domain-specific needs of real-world applications, such as scientific literature review, legal case analysis, financial report synthesis and forecasting, medical diagnosis and treatment guidance, etc. These applications

---

[*]Equal contributions.
[†]Correspondence to Muhan Zhang `<muhan@pku.edu.cn>`.

39th Conference on Neural Information Processing Systems (NeurIPS 2025) Track on Datasets and Benchmarks.

Table 1: Comparison of existing long-context benchmarks and LooGLE v2.

| Benchmark | Avg.Len | #QAs. | Long Dependency | Realistic Task | Auto Labeled | Unseen Docs | Robust Evaluation |
|---|---|---|---|---|---|---|---|
| RULER (Hsieh et al., 2024) | - | - | ✗ | ✗ | ✓ | ✗ | ✓ |
| L-Eval (An et al., 2024) | ~8K | 2,043 | ✗ | ✓ | ✗ | ✗ | ✗ |
| BAMBOO (Dong et al., 2024) | ~16K | 2,804 | ✗ | ✗ | ✗ | ✗ | ✗ |
| LooGLE(Li et al., 2024) | ~20K | 6,448 | ✓ | ✗ | ✗ | ✓ | ✗ |
| ∞Bench(Zhang et al., 2024b) | ~200K | 3,946 | ✗ | ✓ | ✗ | ✓ | ✗ |
| Loong(Wang et al., 2024) | ~250K | 1,600 | ✓ | ✓ | ✗ | ✓ | ✗ |
| LongBench v2(Bai et al., 2024b) | ~240K | 503 | ✓ | ✓ | ✗ | ✗ | ✓ |
| LooGLE v2 (**Ours**) | ~250K | 1934 | ✓ | ✓ | ✓ | ✓ | ✓ |

require comprehensive understanding and deeper reasoning over the given long context instead of relying heavily on the pretrained commonsense knowledge and general linguistic skills, which are currently poorly measured. Furthermore, it has been validated and highlighted in the recent work (Li et al., 2024) that it remains a fundamental challenge for current long-context LLMs to solve **long-dependency tasks**, where multiple pieces of evidence across the long context and a holistic understanding of the document are required, in contrast to the short-dependency tasks such as Needle-In-A-Haystack (NIAH) (Kamradt, 2023). It will be even more complex in real-world applications that require reasoning over dispersed information throughout the extended long texts with global relevance and cross-document coherence.

To address the above-mentioned issues, we propose **LooGLE v2**, an advanced benchmark specifically designed to evaluate the real-world long-dependency understanding and reasoning ability of LLMs. Our benchmark has the following advantages:

- **Real-world data sources.** Our benchmark is built entirely from widely used real-world data sources across diverse domains including **law, finance, game, and code**, spanning legal articles and cases, annual reports of different companies, code repositories, and game trajectories played by LLM agents and real users. There are over 500 domain-specific long documents with an average of 256K tokens, many of which exceed 512K or even 1M tokens, which further drives the development of enhanced models towards "true long-context understanding" rather than merely expanding the context window.

- **Domain-specific long-dependency tasks.** Long-dependency tasks (Li et al., 2024) require reasoning over interdependent pieces of evidence scattered throughout the entire long text. Following this idea, we carefully design **10 types** of domain-specific long-dependency tasks for the above-mentioned data sources, resulting in a total of **1934 QA** instances. Moreover, our benchmark captures the practical challenges involved in real-world long context understanding, enabling a more realistic evaluation of LLMs' ability to handle complex and professional tasks rather than simplified retrieval tasks.

- **Scalability in real world.** Both the number of long texts and the QA samples can be easily scaled, benefiting from our proposed automatic data collection and annotation pipeline. It enables periodic updates with fresh real-world data for task generation and avoids data contamination, a severe problem that previous benchmarks often face. In addition, the controllable input lengths and task difficulties make it more flexible for controlled experiments. Finally, all the tasks are formulated as closed-form questions for a robust evaluation.

A comparison between existing long-context benchmarks and LooGLE v2 can be seen in table 1. Our experiments indicate that our benchmark poses substantial challenges to the long-dependency understanding capabilities of LLMs. We hope that LooGLE v2 will advance the frontier of long-context research in LLMs and foster deeper exploration into empowering LLMs with robust long-dependency reasoning ability.

## 2 Related Works

**Long-Context Benchmarks.** To evaluate long-context understanding of language models, several benchmarks have been proposed in recent years. Pioneering and widely cited benchmarks like NIAH (Kamradt, 2023) and its extensions (Song et al., 2025; Hsieh et al., 2024) focus on synthetic

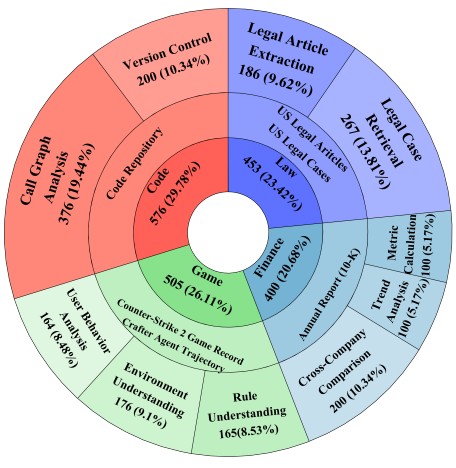
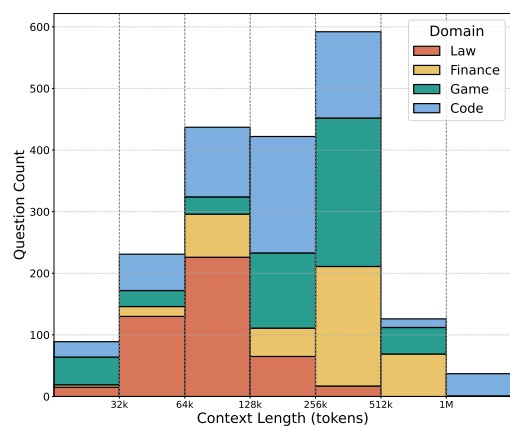

(a) Domain-specific data sources and tasks.     (b) Question context length (in tokens) distributions

Figure 1: Overview dataset statistics for LooGLE v2

retrieval and multi-hop tracing, but mainly targeted shallow reasoning with limited semantic depth and real-world relevance. Most recent comprehensive benchmarks (Shaham et al.; Zhang et al., 2024b; An et al., 2024; Bai et al., 2024a,b; Li et al., 2024; Wang et al., 2024) are designed to cover multiple domains and diverse tasks like multi-document QA, summarization, retrieval and attributing. However, many of them still suffer from key limitations: 1) the context lengths often fall short of modern LLM capacities (Qiu et al., 2024; Bai et al., 2024a), 2) the content is frequently synthetic or stitched, lacking realism (Wu et al., 2024), and 3) the evaluation relies on noisy human annotations (Li et al., 2024; Bai et al., 2024b) or LLM-generated answers with potential bias (Lee et al., 2024; Wang et al., 2024). Moreover, despite task variety, most benchmarks remain centered on retrieval or shallow QA, failing to capture the long-dependency reasoning required in real-world long-context scenarios.

**Long-Context Language Models.** Recent advancements in large language models have substantially extended their context window, with state-of-the-art models (OpenAI, 2024; Reid et al., 2024; Anthropic, 2024; Guo et al., 2025; Liu et al., 2024a) claiming to support up to 128K or even 1M tokens. Meanwhile, various efforts have been made to extend models' context length and enhance their long-dependency capabilities. These include more efficient attention mechanisms (Dao, 2023; Xiao et al., 2024; Yuan et al., 2025), scalable training strategies such as test-time training and parameter-efficient fine-tuning (Sun et al., 2020; Chen et al., 2023), and length-extrapolatable positional encodings (Su et al., 2024; Peng et al.; Ding et al., 2024). Together, these innovations reduce computational overhead while preserving the model's ability to retain distant information, thus enabling more effective reasoning over extended contexts.

## 3 Our Benchmark

In this section, we first introduce the definitions of long dependency tasks in the real world (§ 3.1) for each domain. Then, we present the data curation pipeline, including the source data collection and QA annotation (§ 3.2) which can be reproduced by later works to ensure data scalability.

### 3.1 Task design

**Law**   Legal judgment is one of the most common tasks in legal scenarios. Legal case documents, such as court rulings or judicial decisions, typically follow a structured format comprising several critical sections: case background, cited legal articles, similar case analysis and the judgment outcome. These texts exhibit distinct linguistic and structural features, characterized by formal language, legal jargon, and extremely long content, which is more challenging for LLMs to process and understand effectively. To evaluate the capability of LLMs in legal scenarios, we design the following two tasks:

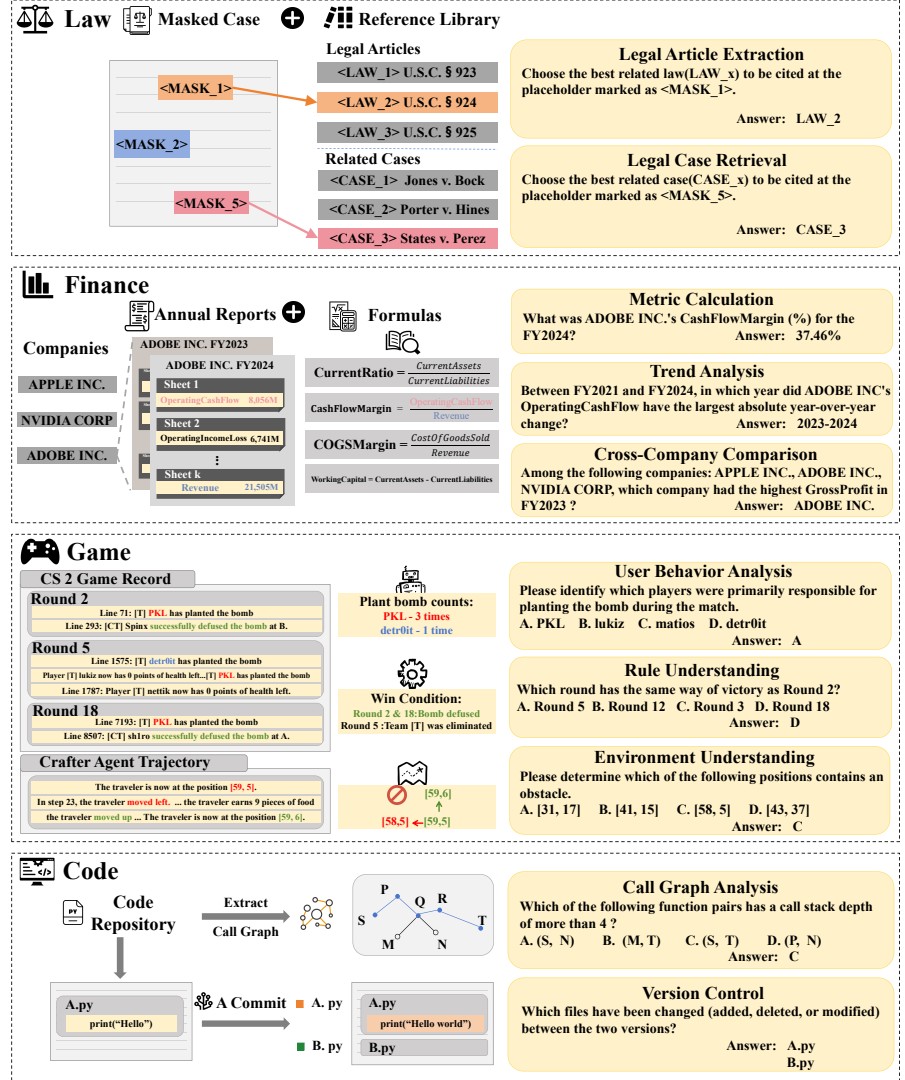

Figure 2: Overview description of LooGLE v2 tasks.

- **Legal Article Extraction** For each legal case document, we first mask the cited legal articles. Then we curate reference articles as a candidate library by mixing the correct articles with a set of nearby distractor articles. The model is asked to select the most appropriate article from the library to fill in each mask. The task relies on long dependency reasoning and comprehension across extracted key entities and concepts. The model must resolve the temporal order of facts and relate them to appropriate legal principles using valid citations from throughout the long text.

- **Legal Case Retrieval** In common law countries, precedential legal cases are cited as evidence to support or challenge the current case. Similarly, the model is asked to complete the masked legal case document by retrieving relevant prior cases as references. Beyond simple keyword matching and semantic similarity, the task also requires LLMs to exhibit advanced long-dependency reasoning. The model must infer implicit fact patterns, identify the core legal issues, verify the consistency of article application, and conduct analogous reasoning to robustly determine whether the cases are substantially similar.

**Finance** Financial statement analysis represents a critical application in finance, where LLMs need to process lengthy, structured documents (typically 50-200 pages per annual reports) that contain both quantitative data and qualitative disclosures. Key analytical tasks in these long documents include financial ratio computation, trend analysis across reporting periods, risk identification, and

comparative benchmarking against industry peers. Performing these tasks effectively requires jointly interpreting tables, notes, and narrative text to retrieve multiple pieces of information, along with applying numerical reasoning to validate calculations against accounting principles. Moreover, temporal reasoning is needed to track quarter-over-quarter or year-over-year changes from the long texts. Below we introduce three representative tasks:

- **Metric Calculation.** Given the long annual report, the task is to precisely extract multiple pieces of numerical information and then compute more complex metrics (like profitability, asset utilization etc.) using provided formula through mathematical reasoning. The process may also involve structured data understanding in various formats like tables or footnotes. This task typically serves as a prerequisite for the subsequent tasks.

- **Trend Analysis.** After the calculation of financial metrics, this task further focuses on evaluating the temporal and causal reasoning capabilities of LLMs. It analyzes the growth or decline trends of a single company through long-term metric comparisons dispersed throughout the extensive text. To arrive at an accurate response, a profound understanding of the question and its correlation with the computed numerical metrics is essential.

- **Cross-Company Comparison.** This task often involves statistical comparisons across different companies, framed as questions that require aggregation or ranking based on quantities, frequencies, durations, specific numbers, and so on. The given inputs consists of extremely long texts that combine reports from multiple companies over several years.

**Game**   In real-world scenarios, textual descriptions of games frequently appear in game commentaries, logs or reports. These descriptions narrate every moment of gameplay including environment settings, player actions, their interactions, and temporal state transitions, ultimately resulting in extremely long contexts. Games usually capture the sequential and causal relationships among different events, users, the changes in environments and roles during the dynamically gameplay. Inferring the rules of game, environment layout, and user behavior patterns solely from the game records or trajectories is particularly important yet challenging. It depends on deeper inductive reasoning rather than merely extracting specific local details.

Therefore, we design three types of tasks to examine how the model deals with these long-dependency information, as detailed below:

- **Environment Understanding** Understanding the in-game environment is a fundamental step toward better performance in gameplay. The model is required to deduce the real embodied environment with spatial awareness of layout, object placement and navigation. Successful completion of the task necessitates jointly stitching together and converting sequences of user actions into a coherent global depiction of the environment across the entire game trajectory.

- **User Behavior Analysis** Understanding the behavior of players provides crucial insight into the progression of the ongoing game and how the players perceive, interpret, and interact with the virtual world. The model must capture key behaviors that directly contribute to score gains or achievements and infer frequent behavior patterns or user preferences from sequences of actions and movements. This task also places greater emphasis on multi-player team coordination for strategy optimization.

- **Rule Understanding** Understanding the rules of games relies on summarization and integration of all the above-mentioned information. It also involves analyzing cooperation among players and interactions between users and the environment with step-wise feedback and rewards. These may help players understand the game's mechanics, objectives, and overall functionality. It is more essential to understand overarching principles rather than grasping the local details of specific entities or events in the game trajectory.

**Code**   Code-related applications represent a key domain for LLM reasoning and have recently received significant attention from both academia and industry. While previous work has primarily focused on tasks such as code generation, execution, and self-repair, our interest lies in more general code comprehension tasks, including workflow analysis and code version management, which are crucial in real-world software development. It is essential to effectively reason over function or class dependencies and understand the temporal evolution of codes for specific projects:

- **Call Graph Analysis** The call graph of a code repository typically represents the inter-dependencies and workflows cross different functions or classes. For each repository, all the code fragments are concatenated file-by-file in a long text. The model is asked to infer the precise call stacks between two selected functions or classes by searching and recovering the intermediate paths based on the function calls that involve multi-hop, cross-module long dependency reasoning across multiple distributed documents.
- **Version Control** The task is widely used in version management to identify code modifications between two commits. The model is required to compare the given two code repositories in long texts, identifying multiple minor local changes over long spans. Then it needs to reorganize these pieces of information as evidence to infer the applied actions such as adding, deleting and replacing, that lead to the changes.

## 3.2 Data Collection and Task Curation

**Law**  We download US legal cases from two major platforms: `CourtListener`[3] and `Westlaw`[4], focusing on cases published after 2024 to minimize the problem of potential data leakage. For each case document, we extract the citation links in `html` and categorize them into relevant articles and cases. For each reference article, we then download all the other articles in the same section to construct the reference article library. Meanwhile, the original documents for reference cases are also collected as the reference case library. We use three types of long legal text: a total of 33 legal case documents, along with their corresponding reference libraries of articles and cases indexed as <LAW_i> or <CASE_i> and replace each cited article or case with placeholder <MASK_i> for QA.

**Finance**  We download 180 10-K annual reports of various companies from year 2020 to 2024 via the `SEC's EDGAR system`[5] as long texts. Based on the financial metrics proposed in Islam et al. (2023), we collect the most frequent financial metrics (in Appendix D) for each year of the companies for further calculation and reasoning. We leverage the tools SEC API and `edgar_sec`[6] to precisely extract the base metrics and filter those derivative metrics that can be calculated using corresponding formulas for the metric calculation task. For the other two tasks, the questions usually ask for the deviations or stability across quarters or years of the same company and statistical metric comparisons among different companies. We adopt a template-based sampling strategy to instantiate these question–answer pairs, ensuring coherent and reliable constructions.

**Game**  We selected Counter-Strike 2[7] (CS2) and Crafter[8] as our data sources. Both games provide rich metadata in long texts for long dependency reasoning. For CS2, we collected 150 game records[9] after 2025. Each record consists of round-by-round player actions (e.g., shooting, equipment usage), game states (e.g., positions, health, scores, kill/death), map layout and bombsite locations. For Crafter, we downloaded 100 trajectories[10] played by LLM agents, containing a task-driven step-wise sequence of the agent's state, actions, positions, resources and achievements in a sandbox environment. Logs and trajectories for both game are in a sequential order and preprocessed into readable texts with semantics based on pre-defined templates (Appendix E) for easier comprehension. Implementation details can be seen in Appendix F.

**Code**  We filter the recent python code repositories in Github from January 2024 to 2025 with fewer than 1000 stars to avoid data contamination, and collect 40 of them with varying lengths. For call graph analysis, we first build the call graph for each repository using the tool `code2flow`[11]. Then we apply depth-first search on the graph and extracted all the call chains with a stack depth between 2 and 5. The model is asked to identify the number of hops between given two functions in the same call chain across multiple files, with at least 2 depths for long-dependency reasoning. For version control, we select six large, actively maintained repositories (e.g. SciPy) and filter commits messages tagged with "fix, bug" to capture meaningful changes. We concatenate the two versions of code

---

[3] https://www.courtlistener.com
[4] https://1.next.westlaw.com/
[5] https://www.sec.gov/edgar
[6] https://github.com/sec-edgar/sec-edgar
[7] https://www.counter-strike.net/cs2
[8] https://github.com/danijar/crafter
[9] https://www.hltv.org/results
[10] https://archive.org/download/crafter_human_dataset
[11] https://github.com/scottrogowski/code2flow

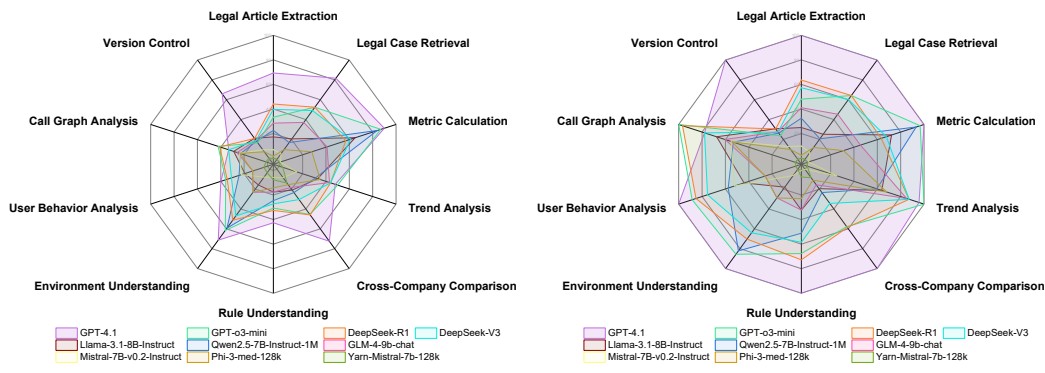

Figure 3: The overall model performance across different tasks on LooGLE v2 (Left: Raw, Right: Normalized to the highest score per task)

before and after the commit for the same code base to form the long texts. For each commit, we use the Git tool to extract git diff information and identify the modified files as the candidate answers.

## 4 Benchmarking LLMs on LooGLE v2

### 4.1 Experimental Setup

**Models** We evaluate our benchmark on 10 representative language models, comprising 6 locally deployed models and 4 API-based models. The locally deployed models include Qwen2.5-7B-Instruct-1M (Yang et al., 2025), LLaMA-3.1-8B-Instruct (Dubey et al., 2024), Mistral-7B-Instruct-v0.2 (Jiang et al., 2023a), GLM-4-9B-Chat-1M-HF (Zeng et al., 2024), Phi-3-Medium-128K-Instruct (Abdin et al., 2024), and Yarn-Mistral-7b-128k (Peng et al.). We also evaluate their larger counterparts with higher parameter sizes (32B,70B), and the complete experimental results are reported in Appendix I. The API-based models include GPT-4.1(Achiam et al., 2023), GPT-o3-mini, DeepSeek-V3(Liu et al., 2024a) and DeepDeek-R1(Guo et al., 2025). These models span a wide range of context window sizes (from 32K to 1M tokens) and parameter scales, providing a comprehensive testbed for long-context evaluation.

**Implement Details** For each domain-specific task, we craft tailored prompt templates to guide the model toward producing structured and consistent responses presented in Appendix B. Based on the conclusion of Liu et al. (2024b), we apply middle truncation for sequences exceeding the model's context window length. To ensure our evaluation is reproducible, we set the model's temperature to 0.1 and top_p to 1.0. More implementation details of our evaluation are shown in Appendix G.
**Metrics** We evaluate the *Version Control* task by computing the Jaccard Similarity (Jaccard, 1901) between the predicted file set and the ground-truth set. For Finance tasks involving numerical short answers, a prediction is considered correct if the relative error is within 5%. A discussion of the rationale for this threshold can be found in Appendix G.4. All other questions are multiple-choice and evaluated by accuracy.

### 4.2 Experimental Results

#### 4.2.1 Main Results

Figure 3 displays models' performance across different tasks and the detailed scores are provided in Table 2. Furthermore, Table 10 reports the performance of models with larger parameter scales. Overall, most models performed poorly across all tasks, indicating that our benchmark poses a substantial challenge spanning diverse real-world scenarios. Among all the models, GPT-4.1 exhibits a substantial advantage over other models on the majority tasks, yet its average score remains at only 59.2%, suggesting ample room for improvement even among the strongest models. Besides, smaller open-source models still lag far behind API-based large models in terms of long-context understanding and task performance. Notably, a longer context window does not necessarily imply

stronger reasoning ability. Despite having a 1M-token context window, GPT-4.1 underperforms the smaller-windowed GPT-o3 (200K) on *Call Graph Analysis* and *Trend Analysis*, both with average input lengths over 200K tokens. This may be due to the multi-hop reasoning and temporal comparison required by these tasks, highlighting that long-span memory alone is insufficient. This observation aligns with our benchmark's core focus on evaluating long dependency understanding.

To further investigate models' long-text capabilities and performance on real-world tasks, we conducted a series of follow-up analyses.

Table 2: Evaluation results of different models across long-dependency tasks (Average).

| Model | | Law | | Finance | | | Game | | | Code | | Avg |
|---|---|---|---|---|---|---|---|---|---|---|---|---|
| Model Name | Window Size | Legal Article Extraction | Legal Case Retrieval | Metric Calculation | Trend Analysis | Cross-Company Comparison | Environmental Understanding | User Behavior Analysis | Rule Understanding | Call Graph Analysis | Version Control | Average |
| **Local-Deployed Models** | | | | | | | | | | | | |
| Llama-3.1-8B-Instruct | 128K | 17.28 | 20.60 | 65.00 | 33.00 | 21.00 | 17.61 | 15.84 | 19.39 | **28.99** | **21.12** | 24.16 |
| Qwen2.5-7B-Instruct-1M | 1M | 22.58 | 16.85 | **84.00** | 31.00 | **27.00** | **24.57** | **59.26** | **24.39** | 23.14 | 18.27 | **28.97** |
| GLM-4-9b-chat | 128K | **28.49** | **37.08** | 41.00 | **43.00** | 18.50 | 17.61 | 23.17 | 12.73 | 26.06 | 19.12 | 25.81 |
| Mistral-7B-v0.2-Instruct | 32K | 6.81 | 5.62 | 1.00 | 15.00 | 8.50 | 5.68 | 7.92 | 22.42 | 23.94 | 9.59 | 11.88 |
| Phi-3-med-128k | 128K | 1.61 | 9.74 | 28.00 | 35.00 | 15.50 | 14.20 | 23.17 | 12.73 | 23.67 | 7.57 | 16.09 |
| Yarn-Mistral-7b-128k | 128K | 0.00 | 0.00 | 0.00 | 3.00 | 14.00 | 7.65 | 3.66 | 3.16 | 1.33 | 1.22 | 3.26 |
| **API-Based Models** | | | | | | | | | | | | |
| GPT-4.1 | 1M | **69.35** | **81.65** | 90.00 | 48.00 | **72.50** | **42.61** | **71.34** | **40.00** | 33.24 | **65.94** | **59.20** |
| GPT-o3-mini | 200K | 33.33 | 52.43 | 87.00 | **50.00** | 45.50 | 30.68 | 61.59 | 35.76 | **42.29** | 16.56 | 43.23 |
| DeepSeek-R1 | 64K | 44.09 | 52.43 | 57.00 | 44.00 | 46.00 | 32.57 | 51.22 | 34.55 | 40.95 | 21.01 | 41.85 |
| DeepSeek-V3 | 64K | 39.78 | 49.06 | 61.00 | 44.00 | 30.50 | 27.27 | 46.95 | 30.91 | 33.24 | 18.97 | 36.71 |

### 4.2.2 Performance on the Varying Input Lengths

To better understand the effect of context length on model behavior, we evaluate the accuracy across several context lengths regimes, where the data is grouped by the length of the input questions in our dataset. Fig. 4 illustrates the trend of model performance with respect to context length. As shown, GPT-4.1 maintains a clear lead, especially beyond 128K tokens. The performance of API-based models drops significantly as length increases, while open-source models remain stably low. This downward trend suggests that current LLMs struggle to fully utilize their context windows. For example, while GPT-4.1 supports a 1M-token context, its effective reasoning window seems to be limited, with a noticeable performance drop beyond 128K tokens. Besides, our long-dependency task design offers a clearer distinction between models: stronger models perform well on shorter contexts but degrade with length, while smaller models consistently show little problem-solving ability regardless of context size.

To further examine whether this degradation stems from context truncation or intrinsic reasoning difficulty, we perform additional ablation studies in Appendix J.1. The results indicate that model performance steadily improves when more intermediate context is retained, implying that models do leverage information distributed throughout long sequences rather than relying solely on the document edges. Moreover, when we decouple reasoning difficulty from length difficulty on the Finance domain (Appendix J.2), the performance gap remains evident—suggesting that long-context reasoning requires both effective memory utilization and genuine multi-step inference capability.

### 4.2.3 Performance with Chain-of-thoughts

To better evaluate the demands of long-dependency reasoning across tasks, we apply chain-of-thought (CoT) prompting to encourage explicit reasoning in model outputs. Following Bai et al. (2024b), we adopt a two-step CoT strategy: the model first generates a reasoning process ("Let's think step by step") and then produces the final answer based on the generated chain of thought. This setup is applied to a randomly selected subset of tasks in LOOGLE v2 to assess its effect on long-range reasoning.

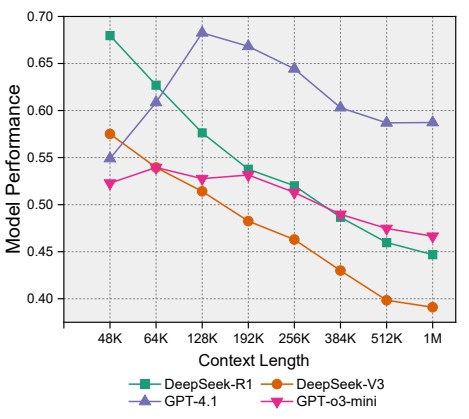

(a) Performance of API-based models.

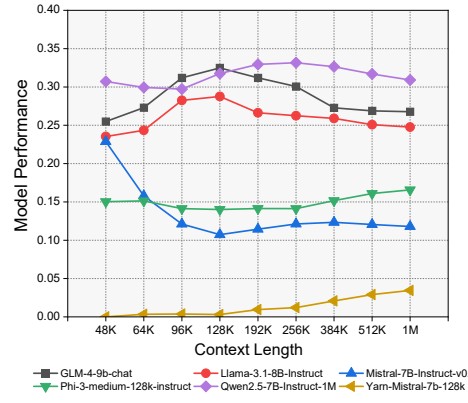

(b) Performance of locally deployed models.

Figure 4: Comparison of model performance across varying input context lengths.

We evaluate CoT prompting on four models, with results summarized in Table 3. Although CoT does not consistently improve overall performance, it benefits tasks requiring structured reasoning, such as *Finance*. Qwen2.5-7B-Instruct-1M and LLaMA-3.1-8B-Instruct show notable gains, GLM-4-9B-Chat remains stable, and Mistral-7B-v0.2-Instruct slightly declines due to its weak baseline.

Table 3: Comparative results of chain-of-thought prompting effects on model performance

| | | Law | | Finance | | | Game | | | Code | | Avg |
| | | | | | | | | | | | | |
| Model | Chain of Thoughts | Legal Article Extraction | Legal Case Retrieval | Metric Calculation | Trend Analysis | Cross-Company Comparison | Environmental Understanding | User Behavior Analysis | Rule Understanding | Call Graph Analysis | Version Control | Average |
| **Locally-Deployed Models** | | | | | | | | | | | | |
| LLaMA-3.1-8B-Instruct | w/o | 17.28 | 20.60 | 65.00 | 33.00 | 21.00 | 17.61 | 15.84 | 19.39 | **28.99** | **21.12** | 24.16 |
| | w/ | 28.14 | 25.22 | 69.67 | 36.00 | **29.00** | 21.02 | 29.07 | **28.08** | 23.49 | 18.9 | 27.94 |
| Qwen2.5-7B-Instruct-1M | w/o | 22.58 | 16.85 | **84.00** | 31.00 | 27.00 | **24.57** | **59.26** | 24.39 | 23.14 | 18.27 | **28.97** |
| | w/ | **32.80** | 35.96 | 82.00 | 33.00 | 26.40 | 14.29 | 46.30 | 16.46 | 21.58 | 12.09 | 28.91 |
| GLM-4-9b-chat | w/o | 28.49 | **37.08** | 41.00 | **43.00** | 18.50 | 17.61 | 23.17 | 12.73 | 26.06 | 19.12 | 25.81 |
| | w/ | 26.70 | 36.96 | 61.67 | 30.00 | 21.17 | 18.18 | 25.81 | 13.74 | 25.97 | 18.07 | 26.54 |
| Mistral-7B-v0.2-Instruct | w/o | 6.81 | 5.62 | 1.00 | 15.00 | 8.50 | 5.68 | 7.92 | 22.42 | 23.94 | 9.59 | 11.88 |
| | w/ | 3.23 | 5.37 | 5.00 | 19.33 | 10.83 | 0.95 | 5.49 | 18.99 | 10.99 | 8.07 | 8.57 |

To further examine how task difficulty scales with reasoning depth, we perform a study on multi-step reasoning in the Code domain. As shown in Appendix J.3, as the depth of function call chains increases, model performance consistently declines—confirming that multi-hop reasoning difficulty exhibits a progressive nature beyond long-context understanding.

### 4.2.4 Performance on Domain-Specific Models

Clear performance gaps emerge when we take a domain-specific view of the results in Table 2. Most models perform poorly, especially on *Legal Article Extraction*, where only GPT-4.1 achieves a decent score (69.35%), revealing the difficulty of retrieving precise clauses from large legal corpora. In finance, performance varies more distinctly: API-based models handle single-metric extraction well but struggle with tasks requiring multi-file or temporal reasoning. Some open-source models fail at metric extraction but score on comparisons mainly by guessing from limited options, not true long dependency reasoning. In the game domain, tasks involving environment and rule understanding require much more challenging long-range reasoning than simpler behavior analysis.

To better imply the impact of domain- and task-specific characteristics on model behavior, we evaluate domain-specific models in the Code and Finance domains and compare their performance with that of general-purpose models on our benchmark in Table 4a and Table 4b.

Table 4: Comparison with domain-specific models across long-dependency tasks.

| Name | Win | Call | Vers | Avg |
|------|-----|------|------|-----|
| CodeLlama-7b | 128K | 26.06 | 5.52 | 18.88 |
| Qwen2.5-Coder-7B | 128K | 30.85 | 21.97 | 27.77 |
| Llama-3.1-8B | 128K | 28.99 | 21.12 | 26.26 |
| Qwen2.5-7B | 1M | 23.14 | 18.27 | 21.45 |
| GLM-4-9b | 128K | 26.06 | 19.12 | 23.65 |

(a) Model comparison on Code

| Name | Win | Calc | Trend | Comp | Avg |
|------|-----|------|-------|------|-----|
| Fin-R1 | 128K | 14.00 | 4.00 | 2.50 | 5.75 |
| Finance-Llama3-8B | 128K | 0.00 | 0.00 | 0.00 | 0.00 |
| Llama-3.1-8B | 128K | 65.00 | 33.00 | 21.00 | 35.00 |
| Qwen2.5-7B | 1M | 84.00 | 31.00 | 27.00 | 42.25 |
| GLM-4-9b | 128K | 41.00 | 43.00 | 18.50 | 30.25 |

(b) Model comparison on Finance

### 4.2.5 Performance with Retrieve Based Methods

To further investigate whether the retrieval-based method significantly improves model performance on our benchmark, we implement the Retrieval-Augmented Generation (RAG) approach for selected models. Details of the implementation can be seen in appendix H.

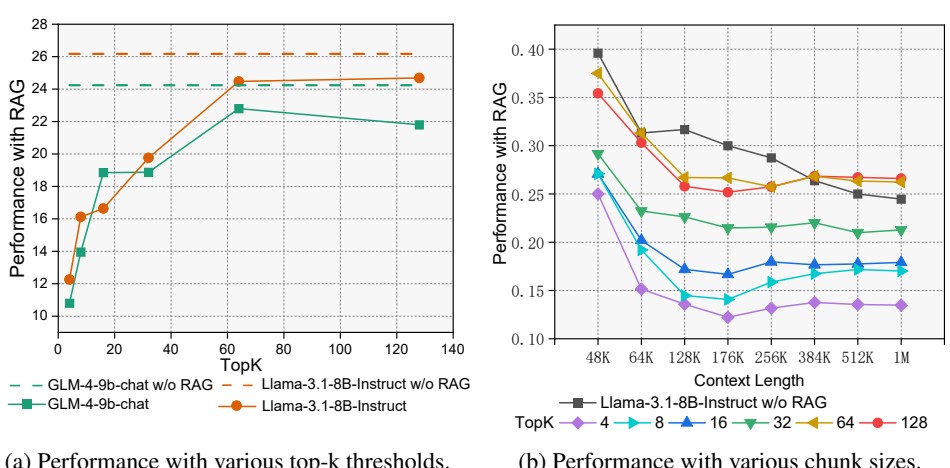

(a) Performance with various top-k thresholds.

(b) Performance with various chunk sizes.

Figure 5: Comparison of model performance using RAG.

The results are presented in Fig.5 and Appendix H, where we observe that the retrieval-based method generally results in a decline in model performance on our benchmark. As shown in Fig.5a, model performance tends to degrade as the number of top-k retrieved chunks decreases. Furthermore, Fig.5b demonstrates that this trend holds across most input context lengths, except in scenarios involving extremely long contexts (above 256K tokens) combined with a relatively large number of retrieved chunks (e.g., k = 128 or 64). This exception can be attributed to the fact that, under extremely long input conditions, models are likely to lose critical information due to middle truncation.

## 5 Conclusions

In this work, we present **LooGLE v2**, a comprehensive and scalable benchmark designed to rigorously evaluate the long-context understanding and reasoning capabilities of large language models on real-world, domain-specific tasks. LooGLE v2 poses a significant challenge for current LLMs, with the best model scoring only 59.2%. Our benchmark provides an essential step towards bridging the gap between extended context window sizes and practical long-context comprehension required for real-world applications.

## 6 Acknowledgements

This work is supported by the National Key R&D Program of China (2022ZD0160300), National Natural Science Foundation of China (62276003), Center of Excellence, Peking University, and CCFTencent Rhino-Bird Open Research Fund.

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

# A   Limitations

We acknowledge several shortcomings in our benchmark construction, which we summarize as follows:

**Limited coverage of real-world scenarios and task design** Due to considerations in question design and the need for reliable benchmark evaluation, our scope is limited to four representative domains with a selected subset of key long-dependency tasks. However, many other real-world long-text scenarios—such as extended debates, policy analyses, and scientific literature reviews—remain unexplored and could provide valuable evaluation challenges.

**Limited length distribution** Although the overall context length in the benchmark is relatively evenly distributed between 16K and 2M tokens, there are notable differences in the average context lengths across different tasks. This may introduce task-specific biases when evaluating model performance across different context lengths. Meanwhile, when assessing the effectiveness of our benchmark through cross-task comparisons of model performance, variations in context length can also act as a confounding factor.

# B   Instructions

This appendix outlines the prompt instructions and input formats used across the 10 tasks in **LooGLE v2**. These prompts are carefully designed to guide models to produce outputs in a consistent format, enabling more robust and reliable evaluation.

## B.1   Law

### B.1.1   Legal article extraction

---

**Instruction:** You are a senior expert in American law. You will now receive a text with two sections:
1. Section <MASKED TARGET CASE> contains an American legal case with several masked placeholders (e.g., <MASK_1>, <MASK_2>). These placeholders represent targeted parts in the case that need to be filled in, while other placeholders (<HIDDEN_INFO>) are included to prevent information leakage.
2. Section <RELATED LAW> provides related legal documents (e.g., <LAW_1>, <LAW_2>, etc.) that may be used as references to fill in those placeholders.

> <text>
> {context}
> </text>

Your task is to analyze the MASKED TARGET CASE and the RELATED LAW, and determine which legal documents (LAW_x) best fills the specific placeholder mentioned in the following question:

> <question>
> {question}
> </question>

You must select only one best answer. Please select the correct option and only response a single sentence with the format as follows:
```
"The correct answer is (LAW_x)"
```

---

### B.1.2 Legal case retrieval

**Instruction:** You are a senior expert in American law. You will now receive a text with two sections:
1. Section <MASKED TARGET CASE> contains an American legal case with several masked placeholders (e.g., <MASK_1>, <MASK_2>). These placeholders represent missing parts in the case that need to be filled in.
2. Section <RELATED CASE> provides related legal cases (e.g., <CASE_1>, <CASE_2>, etc.) that may be used as references to fill in those placeholders.

    <text>
    {context}
    </text>

Your task is to analyze the MASKED TARGET CASE and the RELATED CASE, and determine which related legal case (CASE_x) best completes the specific masked placeholder mentioned in the following question:

    <question>
    {question}
    </question>

You must select only one best answer. Please select the correct option and only respond with a single sentence in the following format:
    `"The correct answer is (CASE_x)"`

### B.2 Finance

All tasks in the Finance domain share a common instruction prompt. Specific requirements for output format and task details are embedded in each individual question. See Appendix C.2 for examples.

**Instruction:** You are a senior financial analyst. You will be provided with certain fiscal year's annual reports for one or more companies. In the provided financial reports, any data enclosed in parentheses () within tables is to be interpreted as a negative value.
The text below contains all the 10-K annual reports in the format <FILE: filename> followed by the file content.

    <text>
    {context}
    </text>

Your task is to analyze the financial information and answer the following question:

    <question>
    {question}
    </question>

### B.3 Game

#### B.3.1 Crafter

**Instruction:** Now, I'm going to present you a textual description of a Crafter game and a question, please read the text and answer the question. Crafter is a survival game in which a traveler explores a jungle, gathering materials and crafting tools to survive. The text I provide shows the traveler's trajectory in this game.
The text is as follows:
    <text>
    {context}
    </text>
After reading the text, please answer the following question:
    <question>
    {question}
    </question>
The options are as follows:
    <option>
    {options}
    </option>
Please select the correct option and only response a single sentence with the format as follows:
    `"The correct answer is (insert answer here)."`

#### B.3.2 Counter-strike 2 (CS2)

**Instruction (User Behavior Analysis):** Now, I'm going to present you a textual description of a Counter-Strike-2 (CS2) game and a question. Please read the text carefully and answer the question. CS2 is a tactical first-person shooter game featuring two teams, each consisting of five players. In the first half of the match, one team plays the role of the Terrorists ([T]), and the other plays the role of the Counter-Terrorists ([CT]). After the first 13 rounds, the two teams switch sides. The goal of the Terrorists is to plant the bomb in a designated area, while the Counter-Terrorists aim to prevent the bomb from being planted or to defuse it after it has been planted. The text I provide shows the gameplay process, including player positions, shooting, utility usage, and bomb plant/defuse actions in this game.
The text is as follows:
    <text>
    {context}
    </text>
After reading the text, please answer the following question:
    <question>
    {question}
    </question>
The options are as follows:
    <option>
    {options}
    </option>
Please select the correct option and only response a single sentence with the format as follows:
    `"The correct answer is (insert answer here)."`

**Instruction (Environment Understanding):** Now, I'm going to present you a textual description of a Counter-Strike-2 (CS2) game and a question. Please read the text carefully and answer the question. CS2 is......(related rules)......After the first 13 rounds, the two teams switch sides. The goal of the Terrorists is to plant the bomb in a designated area, while the Counter-Terrorists aim to prevent the bomb from being planted or to defuse it after it has been planted. In a CS2 match, there are two bomb planting sites (A and B). In each round, the [T] side may plant a bomb at one of the sites. The text I provide shows the gameplay process, including player positions, shooting, utility usage, and bomb plant/defuse actions in this game.

    <text>
    {context}
    </text>

After reading the text, please answer the following question:

    <question>
    {question}
    </question>

The options are as follows:

    <option>
    {options}
    </option>

Please select the correct option and only respond with a single sentence in the following format:

    `"The correct answer is (insert answer here)."`

---

**Instruction (Rule Understanding):** Now, I'm going to present you a textual description of a Counter-Strike-2 (CS2) game and a question. Please read the text carefully and answer the question. CS2 is......(related rules)......After the first 13 rounds, the two teams switch sides. The goal of the Terrorists is to plant the bomb in a designated area, while the Counter-Terrorists aim to prevent the bomb from being planted or to defuse it after it has been planted. The text I provide shows the gameplay process, including player positions, shooting, utility usage, and bomb plant/defuse actions in this game. In the game, at the end of each round, there will be an announcement "The round has ended."

    <text>
    {context}
    </text>

There are five possible victory conditions for each round:
- T side wins when all CT players are eliminated
- T side wins when the bomb is planted and either the countdown or round time expires
- CT side wins when all T players are eliminated
- CT side wins by successfully defusing the bomb
- CT side wins when the round time expires and the bomb was not planted.

After reading the text, please answer the following question:

    <question>
    {question}
    </question>

The options are as follows:

    <option>
    {options}
    </option>

Please select the correct option and only respond with a single sentence in the following format:

    `"The correct answer is (insert answer here)."`

### B.4 Code

#### B.4.1 Call graph analysis

**Instruction:** You are a senior Python expert. You will now receive a full source code of a Python project. The text below contains all the source code in the format '<File>: relative_path\n', followed by the content of each file.

> <text>
> {context}
> </text>

Your task is to analyze the call graph of the project and answer the following multiple choice question.

> <question>
> {question}
> </question>

The options are as follows:

> <option>
> {options}
> </option>

There is only one correct option. You must ensure that your answer is one of the given option letters.
Please select the correct option and only response a single sentence with the format as follows:
> `"The correct answer is (insert answer here)"`

#### B.4.2 Version control

**Instruction:** You are a senior Python expert. You are given two full versions of a codebase, marked by <PREV_VERSION> and <CURR_VERSION>, representing the state before and after a commit.
Each version contains multiple files, and each file begins with a line in the format `<File>: relative_path`, followed by the file content.

> <text>
> {context}
> </text>

Your task is to answer the following question:

> <question>
> {question}
> </question>

Please respond with a list of the relative paths of the changed files, exactly as shown after each `<File>:` in the context.
Please begin your answer with:
> `"The changed files are:[insert relative file paths here]"`

# C  Examples in each domain from LooGLE v2

To facilitate a better understanding of LooGLE v2's domain-specific task design, we present representative questions from each of the 10 sub-tasks in the following section.

## C.1  Law

**Task: Legal Case Retrieval**
**Question:** Choose the best related case(CASE_x) to be cited at the placeholder marked as <MASK_1>.
**Options:** []
**Answer:** CASE_4
**Evidence:**
  original text: 'Porter v. Nussle, 534 U.S. 516, 520, 122 S.Ct. 983, 152 L.Ed.2d 12 (2002)'

**Task: Legal Article Extraction**
**Question:** Choose the best related law (LAW_x) to be cited at the placeholder marked as <MASK_1>.
**Options:** []
**Answer:** LAW_3
**Evidence:**
  original text: '42 U.S.C. 1983'

## C.2  Finance

**Task: Metric Calculation**
**Question:** Based on the annual reports of ELI LILLY AND COMPANY, what was ELI LILLY AND COMPANY's QuickRatio for the FY2020? Please format your response as: "The correct answer is X.XX" (ratio, rounded to 2 decimal places e.g. 1.25). Formula: QuickRatio = (CurrentAssets - InventoryNet) / CurrentLiabilities
**Options:** []
**Answer:** 1.08
**Evidence:**
  CurrentAssets (2020): 17,462,100,000.00
  InventoryNet (2020): 3,980,300,000.00
  CurrentLiabilities (2020): 12,481,600,000.00
  QuickRatio (2020): 1.08

**Task: Trend Analysis**
**Question:** Between FY2020 and FY2024, in which year did COSTCO WHOLESALE CORP /NEW's CapitalExpenditure have the largest absolute year-over-year change? Please format your response as: "The correct answer is XXXX-XXXX" (year pair e.g. 2000-2001).
**Options:** []
**Answer:** 2020-2021
**Evidence:**
  2020-2021: +27.69%
  2021-2022: +8.44%
  2022-2023: +11.10%
  2023-2024: +8.95%

**Task: Cross-Company Comparison**
**Question:** Among the following companies: 1. ELI LILLY AND COMPANY, 2. Exxon Mobil Corporation, 3. Intuitive Surgical, Inc., 4. COCA COLA CO, 5. Abbott Laboratories, which company had the highest WorkingCapital in FY2023? Please answer using the number in front of the company name. Please format your response as: "The correct answer is X" (plain integer number e.g. 2). Formula: WorkingCapital = CurrentAssets - CurrentLiabilities
**Options:** []
**Answer:** 2
**Evidence:**
*WorkingCapital (2023)*:
    Abbott Laboratories: 8,829,000,000.0
    COCA COLA CO: 3,161,000,000.0
    ELI LILLY AND COMPANY: -1,566,200,000.0
    Exxon Mobil Corporation: 31,293,000,000.0
    Intuitive Surgical, Inc.: 6,229,300,000.0

## C.3 Game

**Task: User Behavior Analysis**
**Question:** Usually, the player who planted the bomb the most times is the one responsible for planting the bomb. Based on the description, please determine which of the following players was responsible for planting the bomb during the match?
**Options:**
    A. PKL
    B. lukiz
    C. matios
    D. xureba
**Answer:** A
**Evidence:**
    Line 6497: [T]lukiz has planted the bomb
    Line 7175: [T]PKL has planted the bomb
    Line 8507: [T]PKL has planted the bomb
    Line 10093: [T]PKL has planted the bomb
    Line 16856: [T]matios has planted the bomb

**Task: Environment Understanding**
**Question:** If the traveler attempts to move in a certain direction but his or her position does not change in the next step, it means that the intended destination is blocked by an obstacle. Please determine which of the following positions contains an obstacle.
**Options:**
    A. [28, 29]
    B. [41, 1]
    C. [30, 32]
    D. [26, 1]
**Answer:** C
**Evidence:**
    [28, 29] : The traveler is now at the position <[28, 29]>.
    [41, 1] : The traveler is now at the position <[41, 1]>.
    [30, 32] : The traveler is now at the position <[30, 33]>...In step 22, the traveler moved left...The traveler is now at the position <[30, 33]>.
    [26, 1] : Not mentioned

**Task: Environment Understanding**
**Question:** Please determine which map is the same as the reference map.
**Options:**
    A. Map option 1
    B. Map option 2
    C. Map option 3
    D. Map option 4
**Answer:** A
**Evidence:**
*(inferred from context)*
    Reference map: train
    Map 1: train
    Map 2: ancient
    Map 3: nuke
    Map 4: anubis

---

**Task: Rule Understanding**
**Question:** Which round has the same victory condition as round 3.
**Options:**
    A. Round 1
    B. Round 13
    C. Round 9
    D. Round 20
**Answer:** C
**Evidence:**
    Line 4361: [CT]naitte took down player [T]xureba with a M4A1. ... The round has ended. Side CT won!
    Line 8003: [CT]naitte took down player [T]detr0it with a M4A1. ... The round has ended. Side CT won!

---

**Task: Rule Understanding**
**Question:** In the context, some steps describe actions as "the traveler did <PLACE-HOLDER>". The possible choices for the placeholder are picking, placing, crafting, or other activities. However, we do not know the exact action the traveler performed. You can infer it by comparing the information of the traveler before and after the action. Please identify at which step the traveler did the same thing as at step 13.
**Options:**
    A. Step 160
    B. Step 131
    C. Step 75
    D. Step 182
**Answer:** C
**Evidence:**
    After doing this action, the traveler has 3 pieces of water, 5 pieces of wood
    In step 13, the traveler did <PLACEHOLDER>
    After doing this action, the traveler has 2 pieces of water, 6 pieces of wood
    After doing this action, the traveler has 5 pieces of water, 8 pieces of wood
    In step 75, the traveler did <PLACEHOLDER>
    After doing this action, the traveler has 4 pieces of water, 9 pieces of wood

## C.4 Code

> **Task: Call Graph Analysis**
> **Question:** Which of the following function pairs has a call stack depth of **more than 2**? (e.g., if the call trace for the function pair (A, C) is A→B→C, the call stack depth is 3.)
> **Options:**
>   A. (src/core/video_planner.py::VideoPlanner._generate_scene_implementation_single,
>       src/rag/rag_integration.py::RAGIntegration._generate_rag_queries_narration)
>   B. (evaluate.py::process_theorem,
>       mllm_tools/utils.py::_prepare_text_image_inputs)
>   C. (src/core/video_planner.py::VideoPlanner.generate_scene_outline,
>       src/rag/rag_integration.py::RAGIntegration.set_relevant_plugins)
>   D. (src/rag/rag_integration.py::RAGIntegration._generate_rag_queries_storyboard,
>       mllm_tools/utils.py::_prepare_text_inputs)
> **Answer:** B
> **Evidence:**
>   A: src/core/video_planner.py::VideoPlanner._generate_scene_implementation_single
>         → src/rag/rag_integration.py::RAGIntegration._generate_rag_queries_narration
>   B: evaluate.py::process_theorem
>         → eval_suite/image_utils.py::evaluate_sampled_images
>         → mllm_tools/utils.py::_prepare_text_image_inputs
>   C: src/core/video_planner.py::VideoPlanner.generate_scene_outline
>         → src/rag/rag_integration.py::RAGIntegration.set_relevant_plugins
>   D: src/rag/rag_integration.py::RAGIntegration._generate_rag_queries_storyboard
>         → mllm_tools/utils.py::_prepare_text_inputs

> **Task: Version Control**
> **Question:** Which files have been changed (added, deleted, or modified) between the two versions?
> **Options:** []
> **Answer:**
>   'src/transformers/modeling_utils.py', 'src/transformers/models/blip_2/modeling_blip_2.py',
>   'tests/models/instructblip/test_modeling_instructblip.py'
> **Evidence:**
>   diff --git a/src/transformers/modeling_utils.py b/src/transformers/modeling_utils.py
> (*too long, omitted...*)

## D   Finance metrics

This part presents the base and derivative financial metrics used in the design of our finance-related tasks. These metrics are selected and extended based on the set proposed in (Islam et al., 2023), with refinements to include additional commonly used financial indicators. A detailed list of the metrics used is shown in Table 5.

## E   Text conversion templates and examples for game domain

In this section, we present some examples of templates used for generating textual description documents from game-source data. These templates are designed to accurately convey gameplay information while closely resembling natural language. The raw data obtained from the source is structured, consisting of multiple records, each corresponding to a specific type of event (referred to as 'Event' below). We convert these structured information into textual descriptions based on predefined templates (referred to as 'Template' below) associated with each event type. Detailed method for generating textual descriptions is detailed in Appendix F.

Table 5: Financial metrics used in the finance task

| Base Metrics | | Derivative Metrics | |
|---|---|---|---|
| Revenue | Gross Profit | Gross Margin (%) | Net Profit Margin (%) |
| Operating Expenses | Net Income | Return on Assets (%) | Asset Turnover Ratio |
| Interest Expense | Income Tax Expense / Benefit | Fixed Asset Turnover Ratio | Inventory Turnover Ratio |
| Cost of Goods Sold | Depreciation and Amortization | Accounts Receivable Turnover | Current Ratio |
| Operating Income / Loss | Total Assets | Quick Ratio | Operating Cash Flow Ratio |
| Current Assets | Current Liabilities | Free Cash Flow Growth Rate (%) | Cash Flow Margin (%) |
| Long-Term Debt | Inventory (Net) | Cash-to-Revenue Ratio | COGS Margin (%) |
| Accounts Receivable (Net) | Property, Plant and Equipment (Net) | D&A Margin (%) | Operating Income Margin (%) |
| Operating Cash Flow | Investing Cash Flow | CapEx as % of Revenue | Net Working Capital |
| Capital Expenditure | Cash and Cash Equivalents | Free Cash Flow | EBITDA |
| Marketable Securities | Accounts Payable | Working Capital | |
| Dividends Paid | Shareholders' Equity | | |
| Total Liabilities | | | |

## E.1 Counter-strike 2

**Event:** Killing
**Template:** Oops! Player <killer> took down player <victim> with a <weapon>. It seems the game is so easy for player <killer>. It was/wasn't a headshot! It was/wasn't a wallbang!

**Event:** End of a round
**Template:** The round has ended. Side <winner> won! The score is now T: <t_score> - CT: <ct_score>.

**Event:** Chat message
**Template:** Player <sender> sent an message. The content of the message is '<text>'

**Event:** Player Hurt
**Template:** Oops! Player <attacker> dealt <damage> points of damage to player <victim> with a <weapon>. Player <victim> now has <hp_after> points of health left.

**Event:** Bomb Plant
**Template:** <thrower> has planted the bomb at <bomb_site>. The clock is ticking!

**Event:** Bomb Defuse
**Template:** Fantastic! <thrower> successfully defused the bomb at <bomb_site>. Crisis averted!

**Event:** Frame Snapshot
**Template:** Let's take a look at the conditions and the position of some players. Player <name> of team <team> now has <health> points of health left. The player's remaining money is <money>. And now we discover that the player walked into <room>. / And the player is still at <room>.

## E.2 Crafter

**Event:** Action Execution
**Template:** In step <step>, the traveler <action>.

**Event:** Position Update
**Template:** The traveler is now at the position <[x, y]>.

**Event:** Inventory Change
**Template:** After doing this action, the traveler earns <n pieces of resource, ...>. The traveler's health is now <hp> points.

**Event:** Tool Crafting
**Template:** The traveler crafted a <tool name>.

**Event:** Achievement Unlocked
**Template:** The traveler achieved <achievement name>.

## F    Task annotation details for game

In this section, we present the detailed process of generating natural language textual descriptions and automatically annotating questions for the game domain. All questions in this domain are generated through automated procedures. We describe the structure of the original structured data, the process by which this data is transformed into natural language, the format and characteristics of the resulting textual descriptions, the construction of questions based on key information extracted from the text using predefined *Text Conversion Templates* (see Appendix C.3), and the procedure for extracting supporting evidence and identifying the correct answers.

### F.1    Counter-strike 2

In the tasks related to *Counter-Strike 2*, each original document we downloaded corresponds to a replay file of a single match, stored in a binary format. These files chronologically log in-game events and contain comprehensive gameplay traces, including player location data, various event types (e.g., kills, bomb plants, defusals, damage), player-specific information (e.g., health, weapon, armor), as well as match scores and metadata. We extracted key information from these binary original documents using the CS2 Demo Parser[12], converting the data into a structured JSON format. The resulting JSON files preserve the chronological order of events, with each entry corresponding to a single in-game event. However, we selectively retain only those events that are critical to the game's win conditions and indicative of player performance—precisely the aspects we aim to evaluate LLMs on for understanding. Additionally, we convert player location data from raw coordinates to corresponding room names on the game map. Subsequently, each event tuple in the JSON file is transformed into one or more natural language textual descriptions using predefined sentence templates (see Appendix E). These descriptions collectively form the documents used in our benchmark. The question formats for this task are shown in Appendix C.3. The procedures for question generation and evidence annotation are described as follows.

---

[12] https://github.com/markus-wa/demoinfocs-golang/

### F.1.1 Environment understanding

In our benchmark, each document filename contains the name of the map on which the corresponding match was played. We use this as the ground truth to construct the task. Specifically, we select two documents played on the same map—one is used as the reference match mentioned in the question stem, and the other as the correct option. Additionally, we randomly select three other documents played on different maps to serve as incorrect options. The five selected documents are then concatenated to form the context for the question.

### F.1.2 User behavior analysis

This task includes three distinct types of questions. In the following, we describe the generation method for each question type, as well as the corresponding answer and evidence annotation strategies.

For bomb-related tasks, we design three types of questions, each with a distinct generation and annotation strategy.

For questions targeting bomb planting, we extract all occurrences of the sentence pattern "[T]XXXXXX has planted the bomb at X. The clock is ticking!", where XXXXXX denotes the player's name. We then count the number of bomb plants performed by each player. If the difference between the most and least frequent bomb planter is less than 3, the document is discarded. Otherwise, the player with the highest number of bomb plants is selected as the correct answer, while the three players with the lowest frequencies are used as incorrect options.

For questions focusing on bomb defusal, we extract sentences of the form "[CT]XXXXXX successfully defused the bomb at X. Crisis averted!", again identifying player names from the text. We compute the number of defusals per player. If the difference between the most and least frequent defuser is less than 2, we discard the document. If not, the player with the most defusals is chosen as the correct answer, and three players with the fewest defusals are chosen as distractors.

For questions related to bomb plant site preference, we again extract the sentence "[T]XXXXXX has planted the bomb at X. The clock is ticking!", but this time focus on the bomb site X. We count how often each site is used within a document. If the absolute difference in frequency between the two sites is less than 3, the document is excluded. Otherwise, the more frequently used site is set as the correct answer.

### F.1.3 Rule understanding

From each document, we extract one of the following five sentence patterns at the end of each round to identify the round's win condition:

1. "Fantastic! [X]XXXXXX successfully defused the bomb at X. Crisis averted!
   The round has ended. Side CT won! The score is now T: X - CT: X."

2. "Oops! Player [X]XXXXXX took down player [X]XXXXXX with a XXXX. It seems the game is so easy for player [X]XXXXXX. It wasn't a headshot. It wasn't a wallbang.
   The round has ended. Side CT won! The score is now T: X - CT: X."

3. "The round has ended. Side CT won! The score is now T: X - CT: X." (but not the above two types)

4. "Oops! Player [X]XXXXXX took down player [X]XXXXXX with a XXXX. It seems the game is so easy for player [X]XXXXXX. It wasn't a headshot. It wasn't a wallbang.
   The round has ended. Side T won! The score is now T: X - CT: X."

5. "The round has ended. Side T won! The score is now T: X - CT: X." (but not the fourth type)

Each of these patterns corresponds to a specific type of win condition. For every document, we construct a list that records the win condition of each round in sequential order based on the extracted sentence patterns.

To construct the question, we select two rounds that are sufficiently far apart and share the same win condition. One of these rounds serves as the reference round in the question stem, and the other as the correct option. We then randomly select three additional rounds from the beginning, middle, and end of the list, each with a different win condition, to serve as incorrect options.

### F.2 Crafter

In the tasks related to *Crafter*, each original document we downloaded contains a snapshot of every step in the gameplay. These documents are in JSON format, where each entry records the information of a single step, including the action taken by the player, the player's position, various player attributes (e.g., food, water, energy, materials, and the amount of different tools), and whether the player has completed each achievement up to that point. We first organize this information into a simplified form by computing the stepwise deltas in tool quantities and achievement completion status. Then, we convert the structured data into natural language textual descriptions using predefined sentence templates. These descriptions collectively constitute the documents used in our benchmark. The question formats for this task are shown in Appendix C.3. The procedures for question generation and evidence annotation are described as follows.

#### F.2.1 Environment understanding

From each document, we extract all instances of the sentence pattern: "... The traveler is now at the position [X, Y] ... In step 38, the traveler moved <A DIRECTION>.
The traveler is now at the position [X, Y] ..." where the two position coordinates [X, Y] are identical. Such patterns indicate that the traveler attempted to move one step in the specified direction but remained in the same position, suggesting that there is an obstacle in the intended direction of movement. We traverse the entire document to collect all such patterns and infer the corresponding blocked grid positions, forming a list of obstacle locations. From this list, we randomly select one position as the correct answer. Then, we choose three additional positions that are not in the list as incorrect options.

#### F.2.2 User behavior analysis

From each document, we extract all instances of the sentence pattern: "In step X, the traveler did nothing." Each occurrence of this sentence indicates that the player was sleeping during step X. We collect all such step numbers into a list. We then search this list for any continuous interval of 50 steps during which the player slept without interruption. From these 50-step sleeping intervals, we extract all possible subintervals of 10 consecutive sleeping steps, forming a new list of candidate intervals. A certain 10-step sleeping interval is randomly selected from this list as the correct option. Three additional 10-step intervals that do not belong to this list are sampled as incorrect options.

#### F.2.3 Rule understanding

From each document, we extract all instances of the sentence pattern: "In step X, the traveler did <PLACEHOLDER>." All such step numbers $X$ are collected into a list of unknown actions. For each step in this list, we further extract the textual descriptions from both the current step and the previous step in the format: "... the traveler earns 9 pieces of food, 9 pieces of drink, 8 pieces of energy, 1 piece of sapling, 1 piece of wood. The traveler's health is now 9 points. In step 40, the traveler did something ... After doing this action, the traveler earns 9 pieces of food, 9 pieces of drink, 8 pieces of energy, 1 piece of sapling, 2 pieces of wood. The traveler's health is now 9 points." We compute the difference in resources and health before and after each unknown action and store the results as a dictionary in the format {step: change vector}. We then search this dictionary for two steps with identical change vectors. One is selected as the reference step in the question stem and the other as the correct option. To generate incorrect options, we select three additional steps from the beginning, middle, and end of the dictionary whose change vectors differ from the correct one. These steps are used as incorrect options.

## G Implementation details

### G.1 Model configurations and hyperparameters

We evaluate our benchmark on 10 representative language models, comprising 6 locally deployed models and 4 API-based models. The locally deployed models include Qwen2.5-7B-Instruct-1M (Yang et al., 2025), LLaMA-3.1-8B-Instruct (Dubey et al., 2024), Mistral-7B-Instruct-v0.2 (Jiang et al., 2023a), GLM-4-9B-Chat-hf (Zeng et al., 2024), Phi-3-Medium-128K-Instruct (Abdin et al.,

2024), and Yarn-Mistral-7b-128k (Peng et al.). The API-based models include GPT-4.1(Achiam et al., 2023), GPT-o3-mini, DeepSeek-V3(Liu et al., 2024a) and DeepDeek-R1(Guo et al., 2025). These models span a wide range of context window sizes(from 32K to 1M tokens) and parameter scales, providing a comprehensive testbed for evaluating long-context understanding. All models are evaluated using a decoding temperature of 0.1, top-$p$ of 1.0, and a generation limit of 512 tokens (`max_new_tokens=512`).

### G.2 Hardwares and resource

All evaluations are conducted on four A100 GPUs, each equipped with 80 GB of memory. During evaluation, we first serve the model locally using the vLLM engine (Kwon et al., 2023), and then query the model responses via the OpenAI-compatible client interface.

### G.3 Evaluation metrics for version control task

To evaluate model performance on the VERSION CONTROL task, we use the Jaccard Similarity (Jaccard, 1901) as the primary metric. This task involves identifying the set of files or paths that have changed between two commits. Let $A$ denote the set of predicted changed paths and $B$ denote the ground-truth set. The Jaccard Similarity is defined as:

$$\text{Jaccard}(A, B) = \frac{|A \cap B|}{|A \cup B|} \tag{1}$$

This metric captures the overlap between the predicted and actual changed files, balancing both precision and recall. A score of 1.0 indicates a perfect match, while 0.0 indicates no overlap. We report the average Jaccard score across all evaluated samples.

### G.4 Value Error analysis for finance tasks

As described in Section 4.1, we use **Value Error** as the evaluation metric in the FINANCE domain for numerical prediction tasks. To account for minor discrepancies caused by numerical precision or rounding, we adopt a $5\%$ tolerance threshold. To further validate the sensitivity and robustness of **Value Error**, we conducted an additional significance analysis by examining the distribution of relative differences between predicted and ground-truth values. As shown in Table 6, the results indicate that Value Error values are either concentrated well within the $5\%$ tolerance range or fall significantly outside of it. This provides empirical support for the chosen threshold and demonstrates that the metric effectively distinguishes between correct and incorrect predictions

Table 6: Distribution of Value Error (relative difference) for predictions in Finance tasks. The errors are concentrated well within 5% or significantly above 30%, validating the reasonableness of the 5% tolerance threshold.

| Model / Tolerance Threshold | 0%–5% | 5%–10% | 10%–20% | 20%–30% | >30% (incl. 'null') |
|---|---|---|---|---|---|
| GPT-o3-mini | 143 | 6 | 13 | 10 | 228 |
| DeepSeek-R1 | 118 | 13 | 15 | 13 | 241 |
| DeepSeek-V3 | 94 | 4 | 8 | 7 | 287 |
| GPT-4.1 | 199 | 0 | 16 | 19 | 4 |
| Yarn-Mistral-7b-128k | 28 | 0 | 3 | 1 | 368 |
| Qwen2.5-7B-Instruct-1M | 104 | 1 | 5 | 6 | 279 |
| Phi-3-medium-128k-instruct | 39 | 1 | 10 | 12 | 338 |
| Mistral-7B-Instruct-v0.2 | 17 | 0 | 3 | 7 | 373 |
| Llama-3.1-8B-Instruct | 77 | 1 | 5 | 10 | 307 |
| GLM-4-9b-chat | 55 | 2 | 8 | 6 | 329 |

## H   Detailed results of retrieve based methods

**Single-turn RAG.** In the evaluation of retrieval-based methods, we employ two base models—LLAMA-3.1-8B-INSTRUCT and GLM-4-9B-CHAT—to assess whether retrieval augmentation

can improve models' long-dependency reasoning capabilities. A subset of 645 questions is selected for testing, with the distribution of question types closely matching that of the full benchmark to ensure representativeness. For each test instance, the context is divided into 512-token chunks. Both the chunks and the corresponding question are tokenized using the LONGCITE-GLM4-9B tokenizer (Zhang et al., 2024a). We then compute the semantic similarity between the question and each chunk, selecting the top-$k$ most relevant chunks as the retrieved context. Experiments are conducted with $k = 128, 64, 32, 16, 8$, and $4$ to evaluate performance under varying retrieval granularity.

Table 7 presents the results of the retrieval-based methods, with 'w/o' indicating the baseline without retrieval augmentation. We observe that retrieval augmentation does not lead to consistent performance gains for either LLAMA-3.1-8B-INSTRUCT or GLM-4-9B-CHAT. This suggests that LOOGLE V2 is well-aligned with the characteristics of long-dependency reasoning tasks, where answering typically requires reasoning over dispersed, non-localized information rather than relying on a few highly relevant chunks. In fact, performance often degrades as fewer top-$k$ chunks are used, highlighting the limitations of local retrieval in capturing global dependencies. However, we note an exception in domain-specific tasks such as *Finance*, where certain questions focus on locating specific indicators. In such cases, a small number of retrieved chunks may suffice to provide the necessary information, and moderate performance gains are observed when using retrieval-based methods.

**Multi-turn RAG.** In addition to single-turn retrieval, recent work has also explored multi-turn retrieval strategies, such as those adopted in GSM-Infinite (Zhou et al., 2025). Following this line of research, we further implemented a **multi-turn RAG** approach based on the FLARE strategy (Jiang et al., 2023b), where the model and retriever interact over multiple rounds with progressively refined candidate sets.

As shown in Table 8, multi-turn RAG brings only modest improvements compared with single-turn RAG under comparable retrieval budgets, and performance under both settings remains consistently lower than the no-RAG baseline. This suggests that even iterative retrieval is insufficient to capture the long-range dependencies required by our benchmark.

Table 7: Evaluation of Retrieval-augmented methods on **LooGLE v2**

| Top-k | Law | | Finance | | | Game | | | Code | | Avg |
|---|---|---|---|---|---|---|---|---|---|---|---|
| | Legal Article Extraction | Legal Case Retrieval | Metric Calculation | Trend Analysis | Cross-Company Comparison | Environmental Understanding | User Behavior Analysis | Rule Understanding | Call Graph Analysis | Version Control | Average |
| **Llama-3.1-8B-Instruct** | | | | | | | | | | | |
| 4 | 1.61 | 5.62 | 8.82 | 9.09 | 5.97 | 16.95 | 20.00 | 20.37 | 23.20 | 3.13 | 12.26 |
| 8 | 4.84 | 8.99 | 11.76 | 12.12 | 10.45 | 22.03 | 23.64 | 29.63 | 25.60 | 5.95 | 16.12 |
| 16 | 8.06 | 10.11 | 26.47 | 15.15 | 11.94 | 18.64 | 30.91 | 18.52 | 23.20 | 6.46 | 16.64 |
| 32 | 9.68 | 17.98 | 44.12 | 15.15 | 22.39 | 22.03 | 32.73 | 25.93 | 16.00 | 8.12 | 19.76 |
| 64 | 9.68 | 15.73 | 70.59 | **42.42** | **25.37** | **27.12** | 30.91 | 29.63 | 20.80 | 11.70 | 24.47 |
| 128 | 8.06 | 15.73 | 64.71 | 36.36 | 22.39 | 23.73 | **34.55** | **33.33** | 25.60 | 12.35 | 24.69 |
| w/o | **25.81** | **19.10** | 61.76 | 36.36 | 17.91 | 22.42 | 25.45 | 22.22 | **30.40** | **18.39** | **26.25** |
| **GLM-4-9b-chat** | | | | | | | | | | | |
| 4 | 1.61 | 7.87 | 2.94 | 0.00 | 0.00 | 10.17 | 12.73 | 22.22 | 27.20 | 2.51 | 10.80 |
| 8 | 3.23 | 14.61 | 8.82 | 12.12 | 5.97 | 11.86 | 9.09 | 25.93 | 27.20 | 5.91 | 13.95 |
| 16 | 4.84 | 16.85 | 23.53 | 21.21 | 13.43 | 11.86 | 23.64 | **31.48** | **30.40** | 6.80 | 18.85 |
| 32 | 8.06 | 26.97 | 38.24 | 18.18 | 14.93 | 6.78 | 21.82 | 22.22 | 24.80 | 7.03 | 18.87 |
| 64 | 6.45 | 28.09 | **47.06** | 30.30 | **25.37** | **15.25** | **27.27** | 22.22 | 25.60 | 10.52 | 22.80 |
| 128 | 14.52 | 25.84 | 35.29 | 39.39 | 22.39 | 11.86 | 25.45 | 18.52 | 23.20 | 12.87 | 21.80 |
| w/o | **27.42** | **37.08** | 32.35 | **48.48** | 17.91 | **15.25** | 16.36 | 29.63 | 27.20 | **17.65** | **26.17** |

**Non-LLM retrieval baselines for law tasks** In addition to LLM-based retrieval methods, we also evaluate traditional information retrieval baselines in the LAW domain, where tasks often rely on semantic matching and precise retrieval from large candidate corpora. Specifically, we include **BM25**(Robertson and Zaragoza, 2009) and **TF-IDF** (Sparck Jones, 1972), both of which compute the similarity between the context surrounding the masked target and the candidate pool (e.g., legal articles or related cases). The top-1 retrieved item is selected as the predicted answer.

Table 8: Performance comparison of different RAG strategies versus the full-context (w/o RAG) setting. The results show that RAG is insufficient to address the long-dependency challenges in LooGLE v2.

| RAG Method | Llama-3.1-8B-Instruct | GLM-4-9b-chat |
|---|---|---|
| w/o RAG (Full Context) | 26.25 | 26.17 |
| top-128 (single-turn) | 24.69 | 21.80 |
| top-64 (multi-turn) | 23.79 | 20.98 |
| top-64 (single-turn) | 24.47 | 22.80 |
| top-32 (multi-turn) | 20.82 | 20.31 |
| top-32 (single-turn) | 19.76 | 18.87 |

Table 9: Performance comparison between LLMs and traditional retrieval methods (BM25, TF-IDF) on Legal domain tasks.

| Task / Model Accuracy | Llama-3.1-8B | GLM-9B-chat | Llama-3.3-72B | Qwen-2.5-72B | GPT-4.1 | BM25 | TF-IDF |
|---|---|---|---|---|---|---|---|
| Legal Case Retrieval | 20.60 | 37.08 | 33.71 | 40.82 | 81.65 | 46.82 | 43.82 |
| Legal Article Extraction | 17.28 | 28.49 | 24.19 | 42.47 | 69.35 | 34.95 | 25.81 |
| **Overall** | **19.20** | **33.55** | **29.80** | **41.50** | **76.60** | **41.94** | **36.42** |

From the results in Table 9, we observe that locally deployed small-scale LLMs perform worse than classical retrieval methods such as BM25 in the LAW domain. In contrast, advanced closed-source models like GPT-4.1 still substantially outperform these retrieval baselines, highlighting their stronger contextual reasoning ability. These findings suggest that retrieval remains indispensable for legal-domain tasks, particularly when models operate under limited capacity. They also point to the potential of hybrid approaches that combine LLMs with strong retrieval modules as a promising direction for future research.

# I   Detailed results of models with larger parameter size

Table 10: Evaluation results of different models with varying parameter sizes

| Model | | Law | | Finance | | | Game | | | Code | | Avg |
|---|---|---|---|---|---|---|---|---|---|---|---|---|
| Model Name | Window Size | Legal Article Extraction | Legal Case Retrieval | Metric Calculation | Trend Analysis | Cross-Company Comparison | Environmental Understanding | User Behavior Analysis | Rule Understanding | Call Graph Analysis | Version Control | Average |
| Llama-3.1-8B-Instruct | 128K | 17.28 | 20.60 | 65.00 | 33.00 | 21.00 | 17.61 | 15.84 | 19.39 | 28.99 | 21.12 | 24.16 |
| Llama-3.1-70B-Instruct | 128K | 19.35 | 32.58 | 49.00 | 45.00 | 19.00 | 25.57 | 36.59 | 23.03 | 31.91 | 21.55 | 29.01 |
| Llama-3.3-70B-Instruct | 128K | 24.19 | 33.71 | 67.00 | 42.00 | 21.00 | 28.41 | 36.59 | 28.48 | 28.19 | 17.92 | 30.24 |
| Qwen2.5-7B-Instruct-1M | 1M | 22.58 | 16.85 | 84.00 | 31.00 | 27.00 | 24.57 | 59.26 | 24.39 | 23.14 | 18.27 | 28.97 |
| Qwen2.5-32B-Instruct | 128K | 33.33 | 37.08 | 78.00 | 37.00 | 22.00 | 28.41 | 62.20 | 16.36 | 31.12 | 30.24 | 34.98 |
| Qwen2.5-72B-Instruct | 128K | 42.47 | 40.82 | 62.00 | 31.00 | 17.00 | 25.00 | 51.22 | 26.67 | 33.78 | 20.27 | 33.84 |
| QwQ-32B | 128K | 44.62 | 46.82 | 71.00 | 24.00 | 11.50 | 13.64 | 18.29 | 1.21 | 17.02 | 13.37 | 24.44 |
| GLM-4-9b-chat | 128K | 28.49 | 37.08 | 41.00 | 43.00 | 18.50 | 17.61 | 23.17 | 12.73 | 26.06 | 19.12 | 25.81 |
| GLM-4-9b-chat-1M | 1M | 19.35 | 37.08 | 46.00 | 37.00 | 23.50 | 22.73 | 45.12 | 23.64 | 25.53 | 19.85 | 28.63 |

# J   Supplementary Experiments

## J.1   Effect of context length truncation

A central question in long-context evaluation is whether models are capable of leveraging intermediate evidence rather than relying solely on the beginning or end of documents. Prior work has highlighted the "lost in the middle" phenomenon Liu et al. (2024b), suggesting that intermediate content is often underutilized.

To examine this issue, we vary the context length from 16k to 128k tokens following the truncation strategy used in Bai et al. (2024b), which retains the initial and final segments while progressively excluding middle content at shorter windows. This design enables us to measure how performance degrades when intermediate context is removed.

As shown in Figure 6, performance steadily improves with longer context windows. For example, LLaMA-3.1-8B-Instruct increases from 16.04 at 16k to 24.16 at 128k tokens. This trend is inconsistent with the hypothesis that models rely only on summaries or edge content; instead, it indicates that intermediate information dispersed across the sequence is actively utilized.

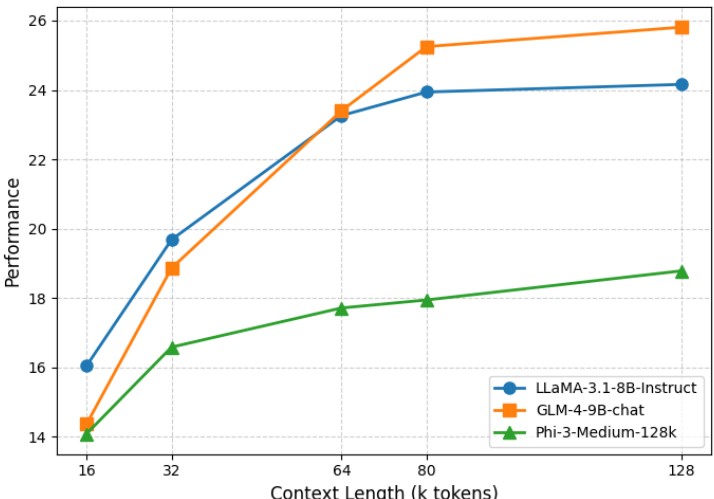

Figure 6: Comparison of model performance across varying input truncation lengths.

## J.2 Decoupling reasoning difficulty from length difficulty

Another important question is whether performance degradation arises primarily from the intrinsic reasoning difficulty of the tasks or from the challenges imposed by long input sequences. To disentangle these two factors, we conduct an ablation on *Finance* tasks. We compare performance under two settings: (i) the *full context* (**w**), where the complete financial reports are provided, and (ii) a *minimal context* (**w/o**), where only the essential values and formulas required to compute the answer are given. This design allows us to separate the effect of long-context processing from the core reasoning challenge inherent to the tasks.

Results are reported in Table 11. Strong models succeed on simple retrieval-style problems, such as extracting a single metric when all necessary values are provided. However, in the majority of tasks, performance drops substantially under the minimal-context setting, showing that success requires more than direct retrieval. These tasks involve reasoning across multiple reports, integrating dispersed evidence, and establishing dependencies across years or companies.

This analysis decouples retrieval difficulty from reasoning difficulty: while larger windows help expose relevant evidence, the core challenge of LooGLE v2 lies in requiring models to identify, connect, and reason over information distributed throughout long contexts.

## J.3 Effect of reasoning depth on model performance

A key aspect of long-context reasoning is the extent to which models can handle tasks requiring multiple inference steps. To verify that multi-hop reasoning difficulty exhibits a progressive nature, we conduct a controlled ablation in the *Code* domain, where task difficulty can be naturally parameterized by the depth of function call chains.

We construct a new test set with call chain depths ranging from 2 to 4, containing 100 instances per level. This design allows us to isolate the effect of reasoning steps while keeping other factors

Table 11: Context study on Finance tasks: full context (w) vs. minimal context with only key information (w/o).

| Finance Task | Mistral-7B | GLM-4-9b-chat | Llama-3.1-8B | DeepSeek-V3 | DeepSeek-R1 | GPT-o3-mini | GPT-4.1 |
|---|---|---|---|---|---|---|---|
| Metric Calculation (w/o) | 56.00 | 72.00 | 52.00 | 100.00 | 100.00 | 100.00 | 100.00 |
| Metric Calculation (w) | 1.00 | 41.00 | 65.00 | 61.00 | 57.00 | 87.00 | 90.00 |
| Trend Analysis (w/o) | 46.00 | 70.00 | 91.00 | 99.00 | 87.00 | 87.00 | 87.00 |
| Trend Analysis (w) | 15.00 | 43.00 | 33.00 | 44.00 | 44.00 | 50.00 | 48.00 |
| Cross-Company Comp. (w/o) | 27.50 | 33.00 | 59.50 | 94.50 | 98.50 | 100.00 | 93.00 |
| Cross-Company Comp. (w) | 8.50 | 18.50 | 21.00 | 30.50 | 46.00 | 45.50 | 72.50 |
| **Avg. Accuracy (w/o)** | **39.25** | **52.00** | **65.50** | **97.00** | **96.00** | **96.75** | **93.25** |
| **Avg. Accuracy (w)** | **8.25** | **30.25** | **35.00** | **41.50** | **48.25** | **57.00** | **70.75** |

Table 12: Model performance in the Code domain across different call chain depths (difficulty levels).

| Model | Difficulty Level (Call Depth) | | |
|---|---|---|---|
| | **2** | **3** | **4** |
| Llama-3.1-8B-Instruct | 31.00 | 29.00 | 23.00 |
| GLM-4-9b-chat | 23.00 | 22.00 | 23.00 |
| Qwen2.5-32B-Instruct | 30.00 | 28.00 | 20.00 |

(such as input length and task format) . We then evaluate several representative models across these difficulty levels.

As shown in Table 12, model performance consistently decreases as the call chain depth increases. This pattern confirms that additional reasoning steps introduce genuine complexity beyond long-context processing alone. The results highlight that multi-hop reasoning constitutes a distinct and progressive challenge within LooGLE v2.

