# OpenReview forum: "LooGLE v2: Are LLMs Ready for Real World Long Dependency Challenges?"
_NeurIPS.cc/2025/Datasets_and_Benchmarks_Track — NeurIPS 2025 Datasets and Benchmarks Track poster_

### Official Review · Reviewer_spc1 · 2025-06-01

**Rating:** 4
**Confidence:** 3

**Summary:**

This work introduces a benchmark targeting the evaluation of long context understanding and retrieving capabilities of LLMs. This dataset features, (1) obtain data from real-world applications, (2) long dependency tasks across domains, and (3) utilization of auto-labeling tasks that can be further scaled up.

The dataset is constructed in the following fields,

(1) Law: tasks include legal artical extraction, legal case retrieval;

(2) Finance: tasks include calculating metrics given information in reports, trend analysis according to the passage, and cross company comparison that requires comparing different finance reports;

(3) Game: tasks include understanding environments; analyzing user behaviors, and understanding rules;

(4) Code: tasks include analyzing calling graphs, comparing two code versions.

After curating such dataset, authors evaluate representative language models, including 6 small (<10B) locally deployed models and 4 API-based models, on subdatasets of each domains, and compared performance against RAG methods.

**Additional Feedback:**

Reference:
[1] GSM-Infinite: How Do Your LLMs Behave over Infinitely Increasing Context Length and Reasoning Complexity? Yang Zhou et al. https://arxiv.org/abs/2502.05252

**Dataset Code Accessibility:**

Yes

**Dataset Code Comments:**

On huggingface the authors have provided all of the problems, and the code provided in the url is about evaluating models on the problems.

**Ethical Considerations:**

No, there are no or only very minor ethics concerns

**Final Justification:**

In the rebuttal period the authors have provided: (1) more explanation and experiment with respect to difficulty levels; (2) more experiments with respect to multi-round RAG; (3) more experiments on open-source models of larger size. They have addressed most of my concerns, hence I raised my score from 3 to 4.

**Limitations Weaknesses:**

(1) The term 'difficulty' is not preciesly defined. I recommend the authors to check this paper[1], in which a metric (reasoning step) is introduced. According to [1], even for the same input length, increasing such reasoning step would increase difficulty, and would be harder for LLMs. This seems to be a better metric for evaluating difficulty of some certain task.

Could you use such metric to classify tasks, and show the difficulty of your QAs?

(2) The RAG method you are using seems to be single-turn RAGs. This, while makes sense, cannot solve simple tasks like 'two needles in a haystack'. Could the authors provide similar experiments with multi-turn RAGs?

(3) The locally deployed models are all small models (<10B). If the authors could provide further results based on large models (e.g. 14B, 32B, 70B, etc.), the benchmarked results would be more complete.

**Strengths Contributions:**

(1) Average Length is long compared to other existing long benchmarks; number of QAs is not small.

(2) Tasks are in real-world cases rather than synthetic cases.

(3) Labels are automatically generated rather than purely human-generated, which are indeed scalable.

(4) The input lengths and task difficulty are claimed to be controllable, thus a robust evaluation is claimed.

---

> ### Author Rebuttal · Authors · 2025-07-31
>
> > Q1. The term 'difficulty' is not precisely defined. I recommend the authors to check this paper[1], in which a metric (reasoning step) is introduced. According to [1], even for the same input length, increasing such reasoning step would increase difficulty, and would be harder for LLMs. This seems to be a better metric for evaluating difficulty of some certain task. Could you use such metric to classify tasks, and show the difficulty of your QAs?
> >
>
> A: Thank you for the thoughtful comment and for pointing us to GSM-Infinite [1]. The idea of using “reasoning steps” as a proxy for difficulty indeed makes much sense in math or logic-based tasks. Nevertheless, in our benchmark, the notion of **difficulty** is defined and controlled in a task-specific manner:
>
> - In **Law**, difficulty is adjusted by varying the **size of the candidate law** article library.
> - In **Finance**, we control the **number of companies or fiscal years** involved in the question.
> - In **Code**, we adjust the **depth of the function call chains** being queried.
> - In **Game**, we vary the **number of game rounds** included in the input to reflect temporal span.
>
> In our design, the **Code** domain aligns most closely with the idea of reasoning steps, as we control task difficulty via the depth of the call chain. Greater depth corresponds to more complex multi-hop reasoning.
> To demonstrate that LooGLE v2 supports **difficulty control**, we constructed a new test set in ***Code*** spanning depths 2 to 4, with 100 instances for each difficulty level, and evaluated the performance of several models across these levels. Partial results are shown below.
>
> | Model/Difficulty | 2 | 3 | 4 |
> | --- | --- | --- | --- |
> | Llama-3.1-8B-Instruct | 31.00 | 29.00 | 23.00 |
> | GLM-4-9b-chat | 23.00 | 22.00 | 23.00 |
> | Qwen2.5-32B-Instruct | 30.00 | 28.00 | 20.00 |
> | Average | 28.00 | 26.33 | 22.00 |
>
> From the results, we observe that as the **difficulty level**—indicated by increasing call stack depth—rises, the models’ performance generally **degrades**, reflecting a decline in their multi-hop reasoning capability.
>
> We appreciate the reviewer's valuable feedback. While the current dataset does not distinguish between difficulty levels to maintain distributional consistency, we will try to incorporate difficulty annotations in future releases, supporting more fine-grained evaluation.
>
> > Q2: The RAG method you are using seems to be single-turn RAGs. This, while makes sense, cannot solve simple tasks like 'two needles in a haystack'. Could the authors provide similar experiments with multi-turn RAGs?
> >
>
> A: We greatly appreciate the reviewers’ comments regarding our RAG methodology. We fully recognize the importance of multi-turn RAG and have conducted additional experiments to address this concern.
>
> We carefully examined the RAG strategies employed in GSM-Infinite [1], particularly *passive RAG* and *interactive RAG*. In our work, we adopted a single-turn RAG approach where the context was split into chunks. For each question, we computed the semantic similarity between the question and each chunk, and retrieved the top-*k* most relevant ones (*k* = 4, 8, 16, 32, 64, 128). The model then answered based solely on these chunks. We considered this method suitable for “multi-needles in a haystack” scenarios, as multiple relevant pieces of evidence could be retrieved simultaneously.
>
> To further explore multi-turn retrieval, we implemented **multi-turn RAG** following the FLARE strategy [2]. We tested two variants: *top-32 3-round* and *top-64 3-round*. In each setting, the model interacted with the retriever over three rounds. For *top-32*, the retriever returned 32 chunks in the first round, then 16 and 8 in the next two rounds based on the model’s follow-up queries. For *top-64*, the chunk counts were 64 → 32 → 16 across the three rounds.
>
> Our results (as shown in the table below) indicate that while multi-turn RAG may yields **modest improvements** over single-turn RAG under the same top-*k* settings, the gains are **limited**. More importantly, model performance under both single-turn and multi-turn RAG remains **significantly lower** than the *no-RAG* setting. This clearly suggests that **even multi-turn RAG is insufficient** to effectively capture the long-range dependencies demanded by our benchmark, further reinforcing its difficulty and the necessity of true long-dependency capabilities.
>
> | RAG Method | Llama-3.1-8B-Instruct | GLM-4-9b-chat |
> | --- | --- | --- |
> | w/o | 26.25 | 26.17 |
> | top-128 | 24.69 | 21.80 |
> | top-64-multi | 23.79 | 20.98 |
> | top-64 | 24.47 | 22.80 |
> | top-32-multi | 20.82 | 20.31 |
> | top-32 | 19.76 | 18.87 |
>
> > Q3: The locally deployed models are all small models (<10B). If the authors could provide further results based on large models (e.g. 14B, 32B, 70B, etc.), the benchmarked results would be more complete.
> >
>
> A:
>
> Thanks for the comment. We agree that including larger-scale models in the local setting provides a more comprehensive view of performance on the LooGLE v2 benchmark.
>
> To address this, we conducted additional experiments on several stronger open-source models: LLaMA-3.1-70B-Instruct, LLaMA-3.3-70B-Instruct, Qwen-2.5-32B-Instruct, and Qwen-2.5-72B-Instruct. Comparative results are shown in the table below.
>
> | Model Name | Window Size | Legal Article Extraction | Legal Case Retrieval | Metric Calculation | Trend Analysis | Cross-Company Comparison | Environmental Understanding | User Behavior Analysis | Rule Understanding | Call Graph Analysis | Version Control | Average |
> | --- | --- | --- | --- | --- | --- | --- | --- | --- | --- | --- | --- | --- |
> | Llama-3.1-8B-Instruct | 128K | 17.28 | 20.60 | 65.00 | 33.00 | 21.00 | 17.61 | 15.84 | 19.39 | 28.99 | 21.12 | 24.16 |
> | Llama-3.1-70B-Instruct | 128K | 19.35 | 32.58 | 49.00 | 45.00 | 19.00 | 25.57 | 36.59 | 23.03 | 31.91 | 21.55 | 29.01 |
> | Llama-3.3-70B-Instruct | 128K | 24.19 | 33.71 | 67.00 | 42.00 | 21.00 | 28.41 | 36.59 | 28.48 | 28.19 | 17.92 | 30.24 |
> | Qwen2.5-7B-Instruct-1M | 1M | 22.58 | 16.85 | 84.00 | 31.00 | 27.00 | 24.57 | 59.26 | 24.39 | 23.14 | 18.27 | 28.97 |
> | Qwen2.5-32B-Instruct | 128K | 33.33 | 37.08 | 78.00 | 37.00 | 22.00 | 28.41 | 62.20 | 16.36 | 31.12 | 30.24 | 34.98 |
> | Qwen2.5-72B-Instruct | 128K | 42.47 | 40.82 | 62.00 | 31.00 | 17.00 | 25.00 | 51.22 | 26.67 | 33.78 | 20.27 | 33.84 |
> | GLM-4-9b-chat | 128K | 28.49 | 37.08 | 41.00 | 43.00 | 18.50 | 17.61 | 23.17 | 12.73 | 26.06 | 19.12 | 25.81 |
> | GLM-4-9b-chat-1M | 1M | 19.35 | 37.08 | 46.00 | 37.00 | 23.50 | 22.73 | 45.12 | 23.64 | 25.53 | 19.85 | 28.63 |
> | GPT-4.1  | 1M  | 69.35 | 81.65 | 90.00 | 48.00 | 72.50 | 42.61 | 71.34 | 40.00 | 33.24 | 65.94 | 59.20 |
>
> As shown in the extended results, larger models generally outperform their smaller counterparts, especially on tasks that require deep comprehension and multi-step reasoning. This trend is particularly clear in the *Legal* domain, where inputs are long, structurally complex, and often require domain-specific interpretation.
>
> Despite the performance improvements, we still observe a **notable gap between these open models and top-tier closed-source models like GPT-4.1**, and all models are still far from solving the tasks, suggesting that long-context reasoning in real-world documents remains a significant challenge even for 70B-scale systems.
>
> We plan to include full results for all large models in the final version and will continue expanding our evaluations.
>
> **Reference**
>
> [1] GSM-Infinite: How Do Your LLMs Behave over Infinitely Increasing Context Length and Reasoning Complexity? Y. Zhou et al.
>
> [2] Active Retrieval Augmented Generation, Z. Jiang et al.

---

> > ### Comment · Reviewer_spc1 · 2025-08-04
> > **Thanks the authors for their response; most of my concers have been addressed**
> >
> > I would like to thank the authors for their response. Most of my concerns have been addressed and I'm happy to raise my score from 3 to 4.

---

> > > ### Author Response · Authors · 2025-08-04
> > >
> > > Thank you for your constructive comments and for raising your score! We sincerely appreciate your recognition of the LooGLE v2 benchmark, and we will include the above details in our revised paper.

---

### Official Review · Reviewer_iciu · 2025-06-14

**Rating:** 4
**Confidence:** 4

**Summary:**

This paper introduces LooGLE v2, a benchmark designed to evaluate large language models (LLMs) on long-context understanding in real-world applications, featuring automatically collected long texts ranging from 16k to 2M tokens across domains like law, finance, gaming, and code. It includes 10 domain-specific long-dependency tasks and 1,934 diverse QA instances to address practical needs.

**Additional Feedback:**

\

**Dataset Code Accessibility:**

Yes

**Dataset Code Comments:**

The code is provided.

**Ethical Comments:**

\

**Ethical Considerations:**

No, there are no or only very minor ethics concerns

**Final Justification:**

my concerns have been fully resolved.

**Limitations Weaknesses:**

1. During data collection, were any filtering processes applied to the data to ensure the factual accuracy of the benchmark?

2. In the legal domain, could non-LLM baseline methods be incorporated into the designed tasks for reference?

3. Although Table 1 mentions that LooGLE v2 uses "unseen docs," the data sources for finance, law, and code domains come from open platforms rather than private data, and the data is not the most recent. How does the paper ensure that the benchmark was not included in the models' training sets?

4. There is a lack of explanation regarding the pipeline process for generating synthetic QA and multiple-choice options in the benchmark.

**Strengths Contributions:**

1. This benchmark includes multiple domains, including law, finance, gaming, and code, which are among the most relevant and popular areas for long-text applications in large language models. This comprehensive coverage ensures that the benchmark is applicable to a wide range of real-world scenarios and provides a robust testbed for evaluating the long-context capabilities of language models across diverse fields.
2. This benchmark constructs domain-specific tasks with long-range dependencies, effectively showcasing their inherent complexity and diversity. These tasks demand that models engage in sophisticated reasoning and demonstrate a profound comprehension of context, moving beyond superficial pattern matching or reliance on general knowledge

---

> ### Author Rebuttal · Authors · 2025-07-31
>
> > Q1. During data collection, were any filtering processes applied to the data to ensure the factual accuracy of the benchmark?
> >
>
> A: Thanks for the insightful question. Yes, we applied a series of filtering and verification steps during data collection and question generation to ensure the reliability and factual correctness of our benchmark.
>
> - First, during data source selection and task design, we strictly adhered to the principle of using **real-world, real-task, non-LLM-generated** data. This ensures that the problems are grounded in authentic information sources, preserving the **factuality and reliability** of the content from the outset.
> - Second, to ensure the correctness of ground truth answers, we applied **multiple traditional (non-LLM) methods** for cross-verification within each task and domain. For every problem, we used at least two independent approaches to extract answers and performed consistency checks. Problems that yielded conflicting results across these methods were discarded. For example, in the *Finance* domain, we extracted metric values both from the source text and via official APIs, and only retained questions where both sources produced consistent results.
> - We also performed **manual spot-checking** on our benchmark to verify the correctness of the evidence extraction and ground truth generation pipeline. Given that our benchmark involves unprecedentedly long contexts, human inspection is highly demanding. Although we sampled a limited number of problems, all reviewed samples passed the verification process. This provides further evidence that our evidence extraction and ground truth generation pipeline is reliable. (**P.S.** This does not contradict the fact that our benchmark is **automatically labeled**. The purpose of manual spot-checking is not to manually label or evaluate individual problems, but rather to verify that our data processing, evidence extraction, and ground truth generation pipeline preserves factuality and reliability throughout.)
>
> > Q2. In the legal domain, could non-LLM baseline methods be incorporated into the designed tasks for reference?
> >
>
> A:
>
> Thank you for the valuable suggestion. We agree that incorporating non-LLM baselines—such as traditional information retrieval methods—can offer meaningful comparisons, particularly in domains like *Law*, where tasks often rely on semantic matching and precise retrieval from large candidate corpora.
>
> To this end, we included **BM25** and **TF-IDF** as non-LLM baselines for evaluating tasks in the legal domain. Specifically, both methods compute the similarity between the context surrounding the masked target and the candidate pool (e.g., legal articles or related cases). The top-1 retrieved item is then selected as the predicted answer.
>
> | Task\Model Accuracy | Llama-3.1-8B-Instruct | GLM-9B-chat | Llama-3.3-72B-Instruct | Qwen-2.5-72B-Instruct | GPT-4.1 | BM25 | TF-IDF |
> | --- | --- | --- | --- | --- | --- | --- | --- |
> | Legal Case Retrieval | 20.60 | 37.08 | 33.71 | 40.82 | 81.65 | 46.82 | 43.82 |
> | Legal Article Extraction | 17.28 | 28.49 | 24.19 | 42.47 | 69.35 | 34.95 | 25.81 |
> | **Overall** | 19.20 | 33.55 | 29.80 | 41.50 | 76.6 | 41.94 | 36.42 |
>
> From the results table, we observe that locally deployed small-scale LLMs perform worse than retrieval-based methods like BM25 on legal domain tasks. In contrast, state-of-the-art closed-source models such as GPT-4.1 still surpass the retrieval baselines by large margins. This disparity may stem from the fact that legal domain tasks are naturally well-suited for retrieval based method — the contextual information surrounding each masked span often provides highly relevant signals for locating the correct content in the corpus.
>
> These findings suggest that **LLM-only** approaches — particularly smaller, locally deployed models—still struggle with effective retrieval in long-context legal tasks. While BM25 serves as a simple retrieval baseline, more advanced retrievers could further enhance performance. Notably, our legal-domain tasks **go beyond standard retrieval** by requiring models to identify and integrate fragmented contextual clues around masked spans. This makes them well-suited for evaluating both contextual understanding and implicit information extraction, and also points to the potential of hybrid approaches that combine LLMs with strong retrieval components.
>
> Thank you again for suggesting the non-LLM baselines. We will include the new results and discussion in the revision.
>
> > Q3. Although Table 1 mentions that LooGLE v2 uses "unseen docs," the data sources for finance, law, and code domains come from open platforms rather than private data, and the data is not the most recent. How does the paper ensure that the benchmark was not included in the models' training sets?
> >
>
> A: Thanks for pointing this out. We sincerely apologize for the confusion caused by the lack of a precise definition of “unseen docs.”
>
> In Table 1’ comparison of different long-context benchmarks, what we intended to convey by “*unseen docs”*  is that the data used in the benchmark are not sourced from any existing benchmark, but are crawled, curated, and labeled from new data sources. All tasks in LooGLE v2 are constructed using a **built-from-scratch pipeline** involving task design, evidence selection, and ground truth generation—ensuring that problems are **newly created and non-overlapping** with prior benchmarks.
>
> While fixed benchmark can hardly completely avoid data leakage as content become public, our **automated pipeline** and **diverse data sources** enable **continuous updates** without manual annotation. For example, domains like *Law* and *Game* regularly produce dozens to hundreds of high-quality long documents daily, making it possible to refresh the benchmark by reapplying our pipeline to newly collected data.
>
> We will revise and further clarify the term “unseen docs” in the final version to avoid any ambiguity. Additionally, we plan to open-source the pipeline framework in future updates.
>
> > Q4. There is a lack of explanation regarding the pipeline process for generating synthetic QA and multiple-choice options in the benchmark.
> >
>
> A: Thank you for raising this point. We clarify that our QA generation pipeline **does not involve LLMs** or any model-generated supervision, in order to maintain full control over the annotation quality and avoid introducing hallucinated content
>
> - For all task types, QAs are generated programmatically based on a set of manually crafted rules and fixed question **templates**, which can be found in **Appendix C**. Both the answers and question-specific info are automatically extracted from the source documents, ensuring that all QA pairs are fully grounded in the original content. For game-related tasks, we first convert raw logs into structured scenario-style representations, then generate questions using rule-based logic. Detailed annotation procedures for these tasks are provided in **Appendix F**.
> - For **multiple-choice options** (mainly in the code and game sections), we first identify the correct answer, and then **sample distractors** from a curated pool of incorrect options. For example, in code-related questions that ask about the call chain depth, distractors are selected from functions with different depths retrieved from the same codebase, ensuring plausibility while maintaining a clear distinction from the correct answer.
>
> We appreciate the suggestion and will include a **more detailed explanation of this process** in the revised version of the paper.

---

> > ### Comment · Reviewer_iciu · 2025-08-02
> >
> > Thank you for your thoughtful rebuttal. I appreciate your clarifications and have decided to raise my score to 4, borderline accept.

---

> > > ### Author Response · Authors · 2025-08-03
> > >
> > > Thank you for your constructive comments and for raising your score! We sincerely appreciate your recognition of the LooGLE v2 benchmark, and we will include the above details in our revised paper.

---

### Official Review · Reviewer_39UX · 2025-06-30

**Rating:** 5
**Confidence:** 4

**Summary:**

The authors propose Loogle V2 a benchmark to evaluate the ability of LLMs to process long contexts to solve a query across diverse domains like code, law, finance, game.

**Dataset Code Accessibility:**

Yes

**Ethical Considerations:**

No, there are no or only very minor ethics concerns

**Final Justification:**

The discussion period did not change my already positive score

**Limitations Weaknesses:**

- Some task setups are questionable: eg. In the game environment for the task of deducing rules, or figuring out the embodied environment, does having just the game commentary suffice in practice even for human testing? It is not very reasonable to me to expect LLMs to be able to do this even if they truly understand long context
- For financial tasks, are the LLMs given access with tools like the mentioned SEC API or metric calculation tools making it an agentic task? If not, there can be a simple ablation study to isolate the reasoning and the long context capturing ability whihc is not clear currently -- if these raw inputs were given to the LLM, is it able to get the answer correctly? Or more analysis is needed, using prompting techniques like CoT to check i fthe model is able to firstly extract the required parts needed for such calculations
- Including an analysis on the performance vs. required context size to solve the task would be helpful to study the trend of models

**Strengths Contributions:**

1. High quality dataset - All of the task environments are drawn from real-world data making it a practical, realistic benchmark. Since most of the domains and tasks are constructed using automeated methods, it is quite scalable and has the ideal property - hard to solve, easy to evaluate
2. Well-organized paper - Paper is well-written and easy to follow
3. Challenging benchmark for current LLMs - although many SoTA LLMs claim to be able to process upto 1M tokens, they do not fare well on tasks requiring this scale of context. Thsi benchamrk would be valuable to the community to measure LLM progress across domains

---

> ### Author Rebuttal · Authors · 2025-07-31
>
> > Q1. Some task setups are questionable: eg. In the game environment for the task of deducing rules, or figuring out the embodied environment, does having just the game commentary suffice in practice even for human testing? It is not very reasonable to me to expect LLMs to be able to do this even if they truly understand long context
> >
>
> A: We gratefully acknowledge the reviewer’s concern. We agree that compared to fully embodied environments, pure text-based game logs are inherently abstract and may not contain all possible details.
>
> To address this, we rewrite the raw logs into scenario-based natural language descriptions, ensuring that each description contains the key reasoning information required by the specific task— such as **action consequences**,  **state transitions**, and **temporal dependencies.** Moreover, the tasks we design do not require highly complex embodied spatial reasoning, but instead focus on understanding agent behavior and environmental cues from the provided context.
>
> We present two illustrative examples below:
>
> **Example 1: Embodied Environment Reasoning**
>
> **Question:** Which location contains an obstacle?
>
> **Context:** A sequence of position and movement logs, where each entry includes the player’s current position and the direction they attempted to move.
>
> **How models solve it:** By comparing the attempted movement direction and resulting position, the model infers whether movement was blocked, indicating the presence of an obstacle.
>
> **Example 2: Rule Deduction**
>
> **Question:** What is the victory condition for a certain game round? (Five possible conditions are listed.)
>
> **Context:** A series of actions, events, and numerical values from the gameplay, such as  player interactions and their states, etc.
>
> **How models solve it:** The model identifies key events and player interactions in the context to determine which victory condition is being satisfied.
>
> > Q2. For financial tasks, are the LLMs given access with tools like the mentioned SEC API or metric calculation tools making it an agentic task? If not, there can be a simple ablation study to isolate the reasoning and the long context capturing ability which is not clear currently -- if these raw inputs were given to the LLM, is it able to get the answer correctly? Or more analysis is needed, using prompting techniques like CoT to check if the model is able to firstly extract the required parts needed for such calculations
> >
>
> A: That’s a really valuable point. In our current setting, LLMs are not equipped with external tools such as SEC APIs or metric calculators and all tasks are completed purely via in-context processing.
>
> We agree that distinguishing the source of difficulty — whether from long-context understanding or the reasoning steps themselves — is an important direction, especially for *Finance* tasks, which often follow an “retrieve-then-reason” structure.
>
> For the reviewer’s reference, we conducted an ablation experiment in financial tasks where the model is **directly provided with the necessary evidence(w/o)**, compared with **full context(w)**.The results are as follows ：
>
> | Task / Setting | Mistral-7B | GLM-4-9b | Llama-3.1-8B | DeepSeek-V3 | DeepSeek-R1 | GPT-o3-mini | GPT-4.1 |
> | --- | --- | --- | --- | --- | --- | --- | --- |
> | **Metric Calculation (w/o )** | 56.00 | 72.00 | 52.00 | 100.00 | 100.00 | 100.00 | 100.00 |
> | **Metric Calculation (w)** | 1.00 | 41.00 | 65.00 | 61.00 | 57.00 | 87.00 | 90.00 |
> | **Trend Analysis (w/o)** | 46.00 | 70.00 | 91.00 | 99.00 | 87.00 | 87.00 | 87.00 |
> | **Trend Analysis (w)** | 15.00 | 43.00 | 33.00 | 44.00 | 44.00 | 50.00 | 48.00 |
> | **Cross-Company (w/o)** | 27.50 | 33.00 | 59.50 | 94.50 | 98.50 | 100.00 | 93.00 |
> | **Cross-Company (w)** | 8.50 | 18.50 | 21.00 | 30.50 | 46.00 | 45.50 | 72.50 |
> | **Avg. Accuracy (w/o)** | 39.25 | 52.00 | 65.50 | 97.00 | 96.00 | 96.75 | 93.25 |
> | **Avg. Accuracy (w)** | 8.25 | 30.25 | 35.00 | 41.5 | 48.25 | 57 | 70.75 |
>
> As shown above, strong models already show promise on **simple retrieval tasks**—such as intuitive fact lookup or metric extraction—when the **evidence is explicit**. In such cases, merely extending the context window can be effective.
>
> While we acknowledge that future analysis using two-step prompting (e.g., CoT or explicit evidence extraction) could yield further insights, it is important to note that these tasks cover only a small portion of our benchmark. The majority of LooGLE v2 tasks **go beyond this retrieve-then-reason paradigm.** They involve deeply intertwined steps of evidence identification, contextual understanding, and multi-step reasoning within extended documents.
>
> > Q3. Including an analysis on the performance vs. required context size to solve the task would be helpful to study the trend of models
> >
>
> A: Thank you for the helpful suggestion. We are also highly interested in understanding how much context is actually needed or effectively utilized by models across different tasks.
>
> We conducted an additional experiment using the truncation strategy described in the paper — by systematically controlling the length of the head and tail portions of the context window. Due to time constraints, we report partial results of **performance vs. required context size** on a subset of models below:
>
> | Model Name/context length | 16k | 32k | 64k | 80k | 128k |
> | --- | --- | --- | --- | --- | --- |
> | Llama-3.1-8B-Instruct | 16.04 | 19.68 | 23.26 | 23.94 | 24.16 |
> | GLM-4-9b-chat | 14.36 | 18.86 | 23.40 | 25.25 | 25.81 |
> | Phi-3-med-128k-instruct | 14.08 | 16.58 | 17.71 | 17.94 | 18.78 |
> | Yarn-Mistral-7b-128k | 3.83 | 3.43 | 4.06 | 5.16 | 3.26 |
> | Mistral-7B-Instruct-v0.2 | 10.34 | 11.29 | - | - | - |
>
> We observe a general upward trend in performance as the context size increases, confirming that longer contexts provide more useful information for many tasks. However, for some models—such as **LLaMA-3.1-8B-Instruct**—the performance gains become marginal or unstable beyond 64k tokens. This suggests that, despite claims of supporting up to 128k context windows, the **effectively utilized** context length may be significantly shorter in practice. Our benchmark thus provides a practical and fine-grained means of evaluating models’ actual ability to leverage long context.

---

### Official Review · Reviewer_4zCC · 2025-07-02

**Rating:** 5
**Confidence:** 4

**Summary:**

This paper introduces LooGLEv2, a dataset for assessing whether language models can accurately establish links and perform specific tasks over long context windows. The dataset has a variety of tasks from legal, financial, gaming, and coding domains. Each one asks the model to answer a question that involves piecing together information over large spans of intervening text. The core finding of the paper is that the best models evaluated are only okay at the task.

**Dataset Code Accessibility:**

Yes

**Dataset Code Comments:**

The dataset is available on Hugging Face: https://huggingface.co/datasets/GraphPKU/LooGLE-v2

The code is available on Github: https://github.com/GraphPKU/LooGLE-v2

**Ethical Considerations:**

No, there are no or only very minor ethics concerns

**Final Justification:**

The author's response to my review was very thorough and provided a lot of new evidence. For this reason, I am raising my score from 4 to 5. Overall, I feel this paper is a solid contribution – the dataset should be very useful for deepening our understanding of how models process long context.

**Limitations Weaknesses:**

The weaknesses I see primarily relate to additional experiments that would help me to further contextualize the results and assess the value of LooGLE itself:

1. I wish we had more insights into whether the intermediate context is being used by model to solve these tasks. The risk of them not using it seems highest for the Finance domain, where the relevant formulae may already be known and so all the models need to do is attend to the initial reports. The legal cases are also a risk here, though the authors took steps to address this by including only post-2024 articles (line 193). Interestingly, the way that the authors deal with context window limitations would provide the needed ablation here, for all models. I would like to see that experiment.

2. How good are models at this task in general, with no distractors? It is hard to tell whether low performance traces to long context or to the inherent difficulty of the tasks. It would seem easy to run the experiments with only the relavant info in the middle. Doing this for at least the strongest model would be illuminating.

3. I would love to see how sensitive the results are to the particular metric decisions given in section 4.1. I am particularly keen to know what happens if the 5% rate for numerical answers is changed, and I would also like to know how these short answers were extracted from model generations. Could there be cases where the model was correct but gave the answer in an unexpected format?

Other notes:

4. Table 1 seems under-described to me. In particular, I am not sure how the checkmark/x distinction is made, especially for things that seem, on the face of it, to be very subjective. What makes the current evaluation "robust" and almost all of the others not robust? Why is "auto labeled" a value? Maybe it would be better to focus on how LooGLE v2 advances LooGLE.

5. I think the Finance panel in Figure 2 has mislabeled the formula in yellow. It should, I suspect, be CashFlowMargin.

**Strengths Contributions:**

This is a creative and interesting new dataset that I think can be used to provide new insights into how models are doing. For this reason, I am basically supportive of publication, though I am eager for more insights into the nature of the benchmark.

---

> ### Author Rebuttal · Authors · 2025-07-31
>
> > Q1: I wish we had more insights into whether the intermediate context is being used by model to solve these tasks. The risk of them not using it seems highest for the Finance domain, where the relevant formulae may already be known and so all the models need to do is attend to the initial reports. The legal cases are also a risk here, though the authors took steps to address this by including only post-2024 articles (line 193). Interestingly, the way that the authors deal with context window limitations would provide the needed ablation here, for all models. I would like to see that experiment.
> >
>
> A:
>
> Thank you for the insightful suggestions. As prior work has shown, models often struggle with the "lost in the middle" issue [1], making it crucial to examine how they utilize intermediate context in long documents.
>
> Since our tasks frequently involve inputs that exceed model context limits, we adopt a **“head+tail” truncation strategy**, which is **commonly used** in other long-context benchmarks [2–4]. While this may omit middle content, it does not substantially simplify the task. Our problems typically require reasoning over **dispersed information**—for instance, in *Finance*, metric calculation often draws from **multiple separate reports**. Unlike standard retrieval tasks, our benchmark tightly couples information extraction with long-context reasoning, making it inherently challenging even when the full context is provided.
>
> To further investigate this, we conducted an **ablation study** by varying the context length from 16k to 128k tokens. As shorter windows progressively exclude more middle content, this setup allows us to evaluate how performance degrades as access to intermediate evidence is reduced.
>
> If models relied solely on head or tail segments (e.g., summaries), performance would plateau early. Instead, we observe that models like LLaMA-3.1-8B-Instruct and GLM-4-9B-chat **consistently improve** with longer contexts—indicating that they benefit from and utilize evidence **dispersed across the entire input**, including middle sections.
>
> | Model Name/context length | 16k | 32k | 64k | 80k | 128k |
> | --- | --- | --- | --- | --- | --- |
> | Llama-3.1-8B-Instruct | 16.04 | 19.68 | 23.26 | 23.94 | 24.16 |
> | GLM-4-9b-chat | 14.36 | 18.86 | 23.40 | 25.25 | 25.81 |
> | Phi-3-med-128k-instruct | 14.08 | 16.58 | 17.71 | 17.94 | 18.78 |
> | Yarn-Mistral-7b-128k | 3.83 | 3.43 | 4.06 | 5.16 | 3.26 |
> | Mistral-7B-Instruct-v0.2 | 10.34 | 11.29 | - | - | - |
>
> > Q2. How good are models at this task in general, with no distractors? It is hard to tell whether low performance traces to long context or to the inherent difficulty of the tasks. It would seem easy to run the experiments with only the relevant info in the middle. Doing this for at least the strongest model would be illuminating.
> >
>
> A:  We’re grateful for this helpful comment. We’d like to clarify that, unlike the single/multi-needle in the hay stack setups where the relevant info can be easily separated from the distracting content, our tasks are intentionally designed to **tightly couple information extraction, reading comprehension, and reasoning with long-context**— which is precisely where the innovation and challenge of our benchmark lies.
>
> For the reviewer’s reference, we conducted an ablation study on *Finance* tasks, which are *Metric Calculation, Trend Analysis and Cross-Company Comparison.* We compared model performance with the **full context** (*w*) versus a **minimal context** (*w/o*), where only the necessary values and formulas were provided. Results are summarized below:
>
> | Finance Task / Models | Mistral-7B | GLM-4-9b-chat | Llama-3.1-8B-Instruct | DeepSeek-V3 | DeepSeek-R1 | GPT-o3-mini | GPT-4.1 |
> | --- | --- | --- | --- | --- | --- | --- | --- |
> | **Metric Calculation (w/o )** | 56.00 | 72.00 | 52.00 | 100.00 | 100.00 | 100.00 | 100.00 |
> | **Metric Calculation (w)** | 1.00 | 41.00 | 65.00 | 61.00 | 57.00 | 87.00 | 90.00 |
> | **Trend Analysis (w/o)** | 46.00 | 70.00 | 91.00 | 99.00 | 87.00 | 87.00 | 87.00 |
> | **Trend Analysis (w)** | 15.00 | 43.00 | 33.00 | 44.00 | 44.00 | 50.00 | 48.00 |
> | **Cross-Company Comparison(w/o)** | 27.50 | 33.00 | 59.50 | 94.50 | 98.50 | 100.00 | 93.00 |
> | **Cross-Company Comparison(w)** | 8.50 | 18.50 | 21.00 | 30.50 | 46.00 | 45.50 | 72.50 |
> | **Avg. Accuracy (w/o)** | 39.25 | 52.00 | 65.50 | 97.00 | 96.00 | 96.75 | 93.25 |
> | **Avg. Accuracy (w)** | 8.25 | 30.25 | 35.00 | 41.5 | 48.25 | 57 | 70.75 |
>
> As shown above, strong models can solve simple retrieval tasks—like metric extraction—when key evidence is explicitly provided. In these cases, increasing the context window may suffice. However, **most tasks in LooGLE v2 require reasoning over long dependencies and integrating dispersed evidence**, where simply extending context length is inadequate. These tasks highlight the need for models to **actively identify, connect, and reason across multiple context segments**, not just passively read longer inputs.
>
> > Q3. I would love to see how sensitive the results are to the particular metric decisions given in section 4.1. I am particularly keen to know what happens if the 5% rate for numerical answers is changed, and I would also like to know how these short answers were extracted from model generations. Could there be cases where the model was correct but gave the answer in an unexpected format?
> >
>
> A: Thanks for the meaningful question. As described in Section 4.1 of our paper, we propose three evaluation metrics: **Jaccard Similarity**, **Value Error**, and **Accuracy**. **Jaccard Similarity** is applied to Version Control tasks in the Code domain, where both predictions and ground truth are sets of changed Python files—making it a natural choice for measuring set overlap. **Value Error** is used in the Finance domain for numerical prediction. To account for possible minor discrepancies due to numerical precision or rounding, we introduce a 5% tolerance threshold.
>
> To address the reviewer’s concern, we conducted additional analysis on the distributions of **Value Error**, defined as the relative difference between the predicted and ground-truth values, to further validate the **sensitivity and robustness** of the metric. The statistical results are shown below.
>
> | Model/Tolerance threshold | 0~5% | 5~10% | 10~20% | 20~30% | >30% (including 'null') |
> | --- | --- | --- | --- | --- | --- |
> | GPT-o3-mini | 143 | 6 | 13 | 10 | 228 |
> | DeepSeek-R1 | 118 | 13 | 15 | 13 | 241 |
> | DeepSeek-V3 | 94 | 4 | 8 | 7 | 287 |
> | GPT-4.1 | 199 | 0 | 1 | 6 | 194 |
> | Yarn-Mistral-7b-128k | 28 | 0 | 3 | 1 | 368 |
> | Qwen2.5-7B-Instruct-1M | 104 | 1 | 5 | 6 | 279 |
> | Phi-3-medium-128k-instruct | 39 | 1 | 10 | 12 | 338 |
> | Mistral-7B-Instruct-v0.2 | 17 | 0 | 3 | 7 | 373 |
> | Llama-3.1-8B-Instruct | 77 | 1 | 5 | 10 | 307 |
> | GLM-4-9b-chat | 55 | 2 | 8 | 6 | 329 |
>
> As illustrated in the tables, the **Value Error** values are either concentrated well within the 5% threshold or significantly outside of it, supporting the reasonableness of this tolerance range. These findings provide empirical evidence that the metric is sensitive and appropriate for evaluating model performance in their respective tasks.
>
> For the **extraction of short answers**, numerical values, and multiple-choice options, we use **regular expression** matching on model outputs to identify answers, which is also widely adopted in other long-context benchmarks [3]. As detailed in **Appendix B**, we provide clear and explicit answer **formatting instructions** to all tasks during inference.
>
> We acknowledge the concern that a model might generate a correct answer in an unexpected format and thus be unfairly penalized. However,  the model's failure on **format adherence** can itself be viewed as a challenge associated with **long-context instruction following**.
>
> > Q4. Table 1 seems under-described to me. In particular, I am not sure how the checkmark/x distinction is made, especially for things that seem, on the face of it, to be very subjective. What makes the current evaluation "robust" and almost all of the others not robust? Why is "auto labeled" a value? Maybe it would be better to focus on how LooGLE v2 advances LooGLE.
> >
>
> A: We apologize for the lack of clarity in the description of Table 1.
>
> The “**robust**”  refers to evaluation setups that do not rely on open-ended generation or LLMs serving as judges—which are often subjective and inconsistent—but instead use objective formats such as multiple-choice or short-answer questions with clearly defined correct answers. This leads to more stable and reproducible evaluation results. Our approach is inspired in part by the conclusion of LooGLE[2] and the design of LongBench v2[3], which also emphasizes format-constrained evaluation to improve robustness.
>
> The “**auto labeled**” column highlights whether a benchmark supports automatic annotation. Many prior datasets, including the original LooGLE, rely heavily on manual labeling, which is time-consuming and can introduce inconsistency or annotator bias. In contrast, LooGLE v2 introduces a  **automated labeling pipeline** that improves annotation consistency, reduces labor, and enables scalable dataset updates over time.
>
> > Q5. I think the Finance panel in Figure 2 has mislabeled the formula in yellow. It should, I suspect, be CashFlowMargin.
> >
>
> A: Thanks for pointing this out. We are sorry for the labeling mistake and will correct it in the final version.
>
> References
>
> [1] *Lost in the Middle: How Language Models Use Long Contexts.*
>
> [2] *LooGLE: Can Long-Context Language Models Understand Long Contexts?*
>
> [3] *LongBench v2: Towards Deeper Understanding and Reasoning on Realistic Long-Context Multitasks.*
>
> [4] *∞Bench: Extending Long Context Evaluation Beyond 100k Tokens.*

---

> > ### Comment · Reviewer_4zCC · 2025-08-03
> > **Score increase**
> >
> > I have raised my overall rating from 4 to 5.

---

> > > ### Author Response · Authors · 2025-08-03
> > >
> > > Thank you for your constructive comments and for raising your score! We sincerely appreciate your recognition of the LooGLE v2 benchmark, and we will include the above details in our revised paper.

---

### Decision · Program_Chairs · 2025-09-18

**Decision:**

Accept (poster)

**Comment:**

This paper introduces LooGLE v2, a benchmark designed to evaluate LLMs on long-context understanding in real-world applications. The benchmark features automatically collected texts ranging from 16k to 2m tokens across domains including law, finance, gaming, and code, with 10 types of domain-specific long-dependency tasks generating 1934 QA instances. The evaluation of 10 representative LLMs shows that even the best-performing model (GPT-4.1) achieves only 59.2% overall accuracy, highlighting significant limitations in current long-context capabilities.

The benchmark clearly addresses a significant gap by using authentic data from diverse, practically relevant domains rather than any synthetic content. Reviewers noted it to be a creative and interesting new dataset that provides realistic challenges spanning legal cases, financial reports, game trajectories, and code repositories. The benchmark also goes beyond simple retrieval tasks to evaluate genuine long-range reasoning across multiple evidence pieces. The 10 tasks require models to integrate information spanning entire documents, addressing the core challenge that reviewers identify as making tasks "hard to solve, easy to evaluate." The evaluation also spans both locally deployed models and API-based models with a thorough analysis across varying lengths and domains.

Reviewers have raised several limitations, including whether models actually use intermediate context. While that is partially addressed through ablation studies, the analysis could be deeper. The authors show performance improvements with longer contexts, but more detailed investigation of which parts of long-documents are being utilized would strengthen their findings. Reviewers also raised valid concerns about the 5% tolerance threshold for numerical answers and potential format-related failures. While the authors provide analysis showing it to be reasonable, reliance on regexes for answer extraction could penalize correct answers in unexpected formats. Another reviewer raised a concern about whether game commentary alone provides sufficient information for environment understanding and rule deduction. While authors have partially addressed that concern, more analysis is needed on that front to address the fundamental question of whether text-based game logs can reasonably support these inference tasks. There are certain other minor issues as well, such as clarity of task descriptions, formalizing the relationship between "difficulty" and task complexity, and better understanding multi-turn approaches regarding the RAG experiments, that should be addressed.

Overall, the methodology is generally sound and the work addresses an important gap in long-context evaluation. The benchmark would be a valuable resource for the community.